## RESEARCH ARTICLE

# A non-transcriptional mitotic function of POU/Oct factors ensures spindle assembly and chromosome segregation

Priya Gohel, Vasilios Tsarouhas, Laveena Kansara, Suresh Sajwan and Ylva Engström*

## ABSTRACT

The POU family, also known as Oct, of transcription factors (POU/Oct), are crucial regulators of cellular processes, including proliferation, cell fate determination, and cancer. Despite their importance, the specific molecular mechanisms by which they influence cell division remain largely unclear. Here, we show that Nub (also known as Pdm1), a *Drosophila* homolog of human POU2F1 (also known as Oct1), is essential for accurate mitotic progression in a non-transcriptional manner. Live imaging and immunostaining in *Drosophila* syncytial embryos reveal that its depletion leads to disorganized spindles, aberrant chromosome segregation and delayed mitotic progression. Similarly, reduction of POU2F1 in live human cells caused disorganized mitotic spindles and spindle collapse. Nub is enriched within the mitotic spindles, and this recruitment is independent of its sequence-specific DNA binding. Instead, it depends on the integrity of spindle microtubules and is regulated by mitosis-related motor proteins, and kinases. Our findings identify both fly Nub and human POU2F1 as important regulators of mitotic progression, acting to maintain spindle stability and proper elongation. The non-transcriptional mitotic role of Nub reveals a previously unrecognized mechanism of POU/Oct proteins and provides new insight into their potential oncogenic properties.

KEY WORDS: Cancer, Cell division, *Drosophila*, Mitosis, Mitotic spindle, Chromosome segregation, Nubbin, Oct1, POU factors

## INTRODUCTION

The POU family, also known as Oct, of transcription factors (POU/Oct) are important players in a variety of cellular processes, including proliferation, cell fate determination, immune function and cancer. It is a versatile group of DNA-binding factors present in all animals (Billin et al., 1991; Bürglin and Affolter, 2016; Herr et al., 1988; McEvilly and Rosenfeld, 1999; Okamoto et al., 1990; Prakash et al., 1992; Rosner et al., 1990; Tang and Engström, 2019; Tantin, 2013; Wegner et al., 1993). Mammalian POU/Oct proteins act as master regulators of pluripotency during embryogenesis, and the pioneer transcription factor POU5F1 (also known as Oct4) is pivotal for the maintenance of embryonic stem cells and for pluripotency in mammals. Together with its partners, POU5F1 has

Department of Molecular Biosciences, The Wenner-Gren Institute, Stockholm University, SE-10691 Stockholm, Sweden.

*Author for correspondence (ylva.engstrom@su.se)

P.G., 0000-0002-0978-5102; V.T., 0000-0002-2933-1351; L.K., 0009-0009-6025-3011; S.S., 0000-0002-6671-5522; Y.E., 0000-0002-8731-4613

the capacity to reprogram somatic cells into induced pluripotent stem cells (Okamoto et al., 1990; Soufi et al., 2012). The related POU2F1 (also known as Oct1) factor is a ubiquitously expressed transcription factor crucial for embryogenesis and neural stem cell specification, and is implicated in a variety of cancers (Vázquez-Arreguín and Tantin, 2016). It serves as the predominant POU/Oct factor in many epithelial tissues, and it has been found to be proto-oncogenic in epithelial malignancies by promoting initiation, progression, and maintenance of epithelial tumors (Jafek et al., 2019; Vázquez-Arreguín et al., 2019).

The *Drosophila* orthologs of the mammalian class II POU factors (POU2F1– POU2F3), called Nubbin (Nub) (also known as POU domain protein 1; Pdm1) and Miti-mere (Miti) (also known as POU domain protein 2; Pdm2), from here on referred to as Nub and Pdm2, are known for their crucial involvement in development and lineage specification. Nub and Pdm2 have been implicated in the regulation of both patterning and of precursor cell division in the CNS, and act as key transcription factors in the temporal cascade of lineage specification during embryonic neurogenesis (Doe, 2017; Gabilondo et al., 2016; Tran and Doe, 2008; Tsuji et al., 2008). In adult gut epithelium regeneration, Nub is involved both in stem cell maintenance and in enterocyte lineage differentiation (Tang et al., 2018). Furthermore, Nub, and to some extent also Pdm2, has been shown to be involved in wing and leg specification (Cifuentes and García-Bellido, 1997; Loker and Mann, 2022; Neumann and Cohen, 1998; Ng et al., 1995; Rauskolb and Irvine, 1999).

The *nub* gene is organized in two major transcription units with independent promoters, and these pre-mRNAs produce two distinct protein variants – Nub-PB (long isoform) and Nub-PD (short isoform) (Fig. S1A). The shared C-terminal region includes the DNA-binding POU domain and homeodomain. It has recently been shown that the two Nub isoforms play antagonistic roles in the regulation of gut immunity, as well as in the control of stemness and differentiation in the regeneration of the gut epithelium (Lindberg et al., 2018; Tang et al., 2018). During *Drosophila* embryogenesis, zygotic *nub* expression is observed as a broad stripe just before cellularization, followed by splitting into two stripes (Affolter et al., 1993; Dick et al., 1991; Lloyd and Sakonju, 1991). Subsequent *nub* expression is dynamic throughout embryogenesis and is especially apparent in early neurogenesis when essentially all cells of the neurogenic region express *nub* (Yang et al., 1993). During later stages of embryogenesis, subsets of neuroblasts of the developing CNS express *nub*, as well as cells in the developing anterior and posterior midgut primordia (Affolter et al., 1993). Nub, and to some extent Pdm2, have been implicated in the regulation of a switch between the continued proliferation of stem cells and lineage specification (Bhat and Apsel, 2004; Doe, 2017; Seroka et al., 2020; Tang et al., 2018). However, how Nub acts to maintain proliferation of neuroblasts and intestinal stem cells is not understood.

There is a wealth of studies pointing to the important roles of POU/Oct proteins in regulation of cell cycle progression linked to

proliferation and also to cancer (Ben-Batalla et al., 2010; Caelles et al., 1995; Castrillo et al., 1991; Jullien et al., 2015; Shin et al., 2016; Zhao et al., 2014), but few have reached a molecular understanding of the underlying mechanisms. The ubiquitously expressed POU2F1 factor is associated with proliferation in cell lines and linked to several cancer types (Jafek et al., 2019; Stepchenko et al., 2022; Vázquez-Arreguín et al., 2019; Vázquez-Arreguín and Tantin, 2016). In addition, POU2F1 has been found to be regulated by phosphorylation and ubiquitylation in a cell cycle-dependent manner, which affects its interaction with chromatin (Kang et al., 2011).

Here, we intended to gain further mechanistic understanding of the role of POU/Oct proteins in cell proliferation by following cell cycle progression through live imaging in *Drosophila* syncytial embryos, and fly and human cell cultures. The well-established pattern of rapid synchronous nuclear divisions in *Drosophila* syncytial embryos, coupled with the absence of transcription during this period (Edgar and Schubiger, 1986; Harrison et al., 2023; Kwasnieski et al., 2019), makes this model particularly well-suited for exploring putative 'non-transcriptional' roles of regulatory proteins. Many transcription and splicing factors end up on mitotic components, such as the midbody, centrosomes, kinetochores and spindles, during cell division (Kang et al., 2011; Neumann et al., 2010; Pellacani et al., 2018; Somma et al., 2020; Yokoyama et al., 2009), but their roles in mitosis remain elusive.

We found that depletion of Nub-PD isoform, but not Nub-PB, leads to mitotic defects in *Drosophila* S2 cells. In early embryos, loss of Nub-PD causes aberrant anaphase spindles with reduced motility, incorrect chromosome segregation and mitotic failure, resulting in a significant loss of cortical nuclei and poor embryo survival. These findings suggest that in syncytial embryos, Nub-PD plays a non-transcriptional role that is crucial for the proper timing and progression of mitosis. Similarly, downregulation of POU2F1 in human cells leads to disorganized mitotic spindles and multinucleated cells. Upon mitosis, Nub-PD is enriched around the mitotic spindles, through a regulated mechanism involving components of the microtubule motor proteins Klp61F and Klp3A, the polarity determinant Crumbs, and the kinases Cdk1, Nek2 and NiKi (also known as Nek9). This localization was independent of its sequence-specific DNA binding but depended on spindle microtubule integrity. Together, these findings highlight the crucial role of both insect and human POU/Oct factors in mitotic progression, offering new insights into their involvement in normal and malignant cell proliferation.

## RESULTS

### Depletion of Nub-PD perturbs mitosis in *Drosophila* S2 cells

In *Drosophila,* the two Nub protein isoforms, Nub-PB and Nub-PD (Fig. S1A), play antagonistic roles in gut epithelium regeneration, with Nub-PB driving differentiation and Nub-PD maintaining stemness (Tang et al., 2018). How Nub protein isoforms control proliferation and/or differentiation in opposite directions and how this couples to the cell cycle is not understood. To address this, we first followed the progression of mitosis after isoform-specific RNA interference (RNAi) (Fig. S1B,C). Upon *nub-RB* RNAi, no mitotic phenotypes were observed. In contrast, *nub-RD* RNAi caused striking mitotic defects, with multipolar spindles (34%) and multinucleated cells (32%) compared to control (*luc-RNAi*) at 5.5% and 7.2%, respectively (Fig. 1A,B). The presence of multinucleated cells after division was further supported by Lamin staining (Fig. S1D), confirming disrupted nuclear organization after mitosis, suggesting incomplete cytokinesis in these cells. Live imaging of dividing S2 cells expressing histone 2B–GFP (H2B–GFP) and α-tubulin–mCherry, revealed significant delays in metaphase to anaphase

transition along with prolonged cytokinesis in *Nub-RD* RNAi cells compared to control cells (Fig. 1C). The delay in mitotic progression was accompanied by defects in spindle integrity in *nub-RD* RNAi cells. Following the cell-cycle phase distribution in fly fluorescent ubiquitination-based cell cycle indicator (FUCCI) S2 cells (Zielke et al., 2014) showed that the proportion of cells in S phase was unaffected by the knockdown of either isoform (Fig. S1E). However, *nub-RD* knockdown significantly increased the proportion of G2/M phase cells and reduced G1 phase cells, indicating a block in mitotic entry or progression. By contrast, *nub-RB* knockdown slightly reduced the proportion of G2/M phase cells and increased G1 phase cells (Fig. S1E), suggesting distinct roles for the two Nub isoforms during the cell cycle. Collectively, these results indicate an important role of Nub protein in cell division, with the entry and/or progression through mitosis specifically requiring the Nub-PD isoform.

### POU2F1 is required for mitotic progression in human cells

To examine whether a mitotic function is evolutionarily conserved among POU/Oct factors, we investigated whether the human homolog POU2F1 had a similar role to that of fly Nub-PD in mitotic progression. siRNA-mediated knockdown of POU2F1 in HeLa cells, combined with live-cell imaging, revealed that POU2F1 depletion caused prolonged mitosis, disorganized spindle structures and cytokinesis failure (Fig. 1D,E). These results were consistent with quantitative experiments in fixed HeLa and HEK293 cells, with a 3.5-fold increase in multinucleated cells (28.2%) compared to controls (6.7%, $P<0.01$; Fig. S1F–G), and a 3.8-fold increase in HEK293 cells (28.6% versus 7.8%, $P<0.01$; Fig. S1H,I). These findings are similar to those seen upon Nub-PD downregulation in *Drosophila* S2 cells (Fig. 1A–C), suggesting an analogous and possibly evolutionarily conserved role of these POU/Oct proteins in mitotic timing and progression.

### Sequence-specific DNA binding is not required for mitotic spindle localization of Nub-PD

To further investigate the mitotic role of Nub-PD, we generated Nub-PD constructs tagged with monomeric Kusabira-Orange 2 (mKO2) and red fluorescent protein (RFP), and analyzed their intracellular localization by live imaging in *Drosophila* S2 cells. Nub-PD–mKO2 and Nub-PD–RFP localized to the nucleus during interphase (Fig. 1G; Fig. S1J,K). During mitosis, however, Nub-PD–mKO2 and Nub-PD–RFP localized around the mitotic spindle (Fig. 1H,I; Fig. S1J), overlapping with tubulin staining, further supporting a mitotic role. Next, we explored whether the localization around the mitotic spindle involved amino acids required for transcriptional regulation or not. The nuclear localization and DNA-binding properties of the POU domain and homeodomain of POU/Oct factors have been extensively studied (Phillips and Luisi, 2000; Sturm and Herr, 1988). Structure–function analyses have pinpointed the amino acids that are crucial for sequence-specific DNA binding and transcriptional regulation (Assa-Munt et al., 1993; Dekker et al., 1993; Klemm et al., 1994; Sturm et al., 1988). The DNA-binding POU specific (POU$_S$) domain and POU homeodomain (POU$_H$) of *Drosophila* Nub and human POU2F1 show 95% and 83% identity in the POU$_S$ and POU$_H$ domains, respectively. Building on this knowledge and high amino acid conservation, we expressed tagged Nub-PD protein variants with mutations in the POU$_S$ and POU$_H$ domains that are identical in the two proteins, and have been shown to be crucial for sequence-specific DNA binding of POU2F1 (Klemm et al., 1994). We mutated the POU$_S$ domain at Q464A, T465A and R469A; the POU$_H$ at V569A, C572A, N573A and Q576A; and the nuclear localization signal (NLS) at R522A and R524A (Fig. 1F). As

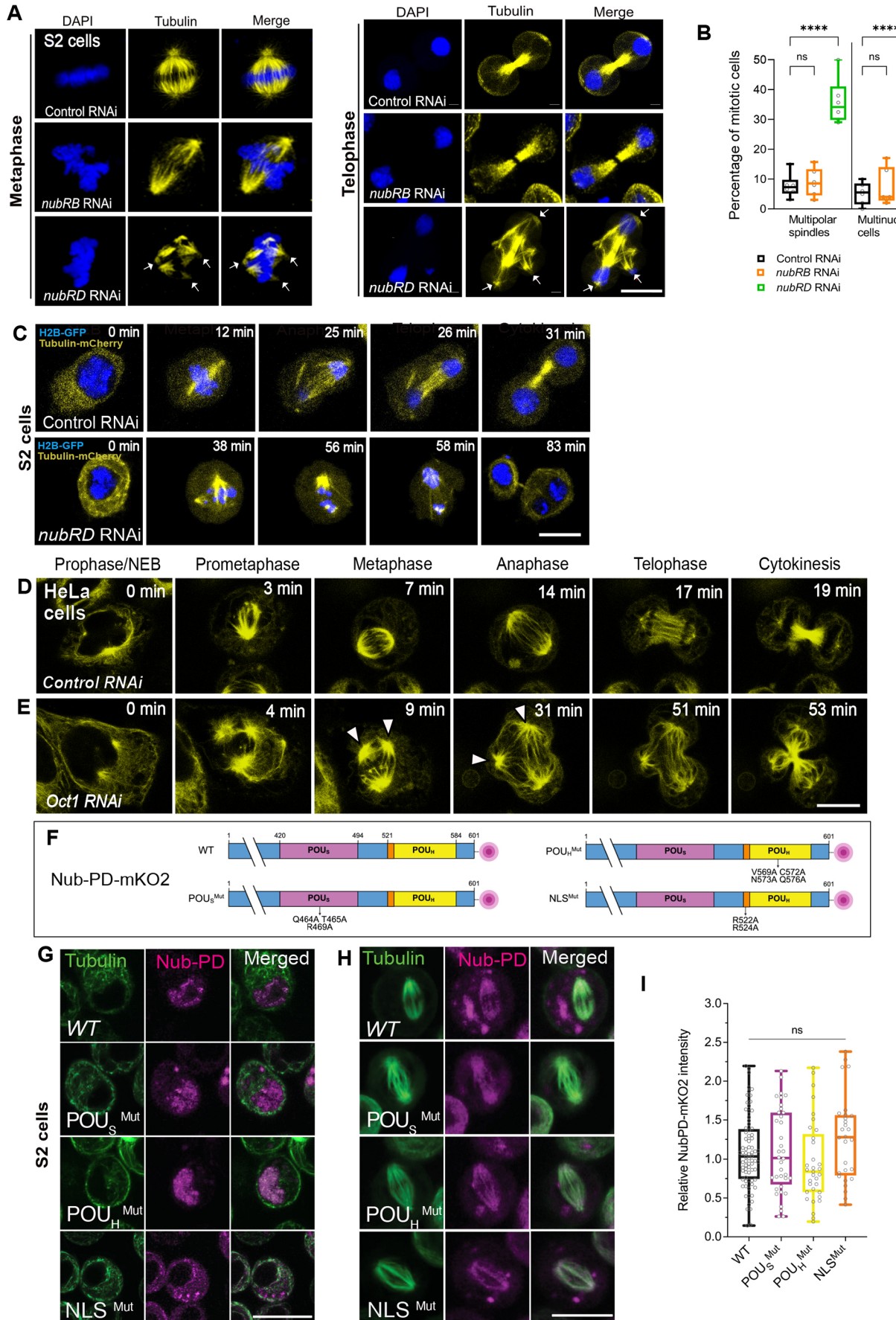

**Fig. 1.** See next page for legend.

**Fig. 1. RNAi-mediated depletion of Nub-PD causes mitotic spindle defects in *Drosophila* S2 cells, independently of its nuclear function.**
(A) Confocal images of mitotic S2 cells in metaphase and telophase, treated with dsRNA as indicated (RNAi) and processed for β-tubulin immunofluorescence (yellow). DNA was counterstained with DAPI (blue). White arrows indicate abnormal spindle morphology. (B) Box and whisker graphs showing mitotic defects (multipolar spindles and multiple nuclei) in control, *nub-RB RNAi*, and *nub-RD* RNAi-treated cells. N=6 biological replicates and n=620 mitotic cells. ****$P$<0.0001; ns, not significant (one-way ANOVA followed by Tukey's multiple comparison test). (C) Representative images from live imaging of S2 cells stably expressing H2B–GFP (blue) and α-tubulin–mCherry (yellow). Mitotic progression of control and *nub-RD* RNAi-treated cells, relative time (min) as indicated. Images representative of five experimental repeats. (D,E) Representative images of live HeLa-cells treated with control siRNA (D) or POU2F1 (Oct1) siRNA (E) and incubated with ViaFluor® 488 dye to visualize the spindle microtubules (yellow). The relative time (min) during mitotic progression is indicated. White arrowheads indicate mitotic spindles with more than two poles (multipolar spindles). Images representative of three experimental repeats. (F) Graphical representation of Nub-PD–mKO2 protein domains of wild type (WT) and mutated Nub-PD variants, tagged in the C-terminus with monomeric mKO2 (red circle). WT, full length Nub-PD–mKO2 protein; POU$_S^{Mut}$, mutated POU specific (POU$_S$) domain at Q464A, T465A and R469A; POU$_H^{Mut}$, mutated POU de-polymerization in (POU$_H$) at V569A, C572A, N573A and Q576A; NLS$^{Mut}$, mutated nuclear localization signal (NLS) at R522A and R524.
(G,H) Confocal images of live S2 cells during interphase (G) and metaphase (H) transfected with the Nub-PD–mKO2 plasmids. Cells were incubated with ViaFluor® 488 dye to visualize the spindle microtubules (green). The mKO2 signals (magenta) are overlapping with microtubules (green) for the WT and all mutated variants. (I) Graph showing the relative spindle intensity of mKO2 in WT and mutated variants of Nub-PD–mKO2. ns, not significant (unpaired two-tailed *t*-tests). The box plots in B and I show the median (horizontal line), and the data range from 25th to 75th percentiles. The bars indicate maximum and minimum values. Scale bars: 10 µm.

expected, mutations in the NLS blocked nuclear localization of Nub-PD–mKO2 in interphase cells (Fig. 1G; Fig. S1K), whereas mutations in the POU domain and homeodomain did not (Fig. 1G). In metaphase cells, both the NLS mutant and the Nub-PD POU domain and homeodomain mutants localized to the mitotic spindle (Fig. 1H,I), indicating that sequence-specific DNA binding is not required for this localization. Thus, sequences in Nub-PD required for nuclear targeting and transcriptional regulation are not crucial for the localization to the mitotic spindle, and therefore unlikely to be involved in the mitotic functions of this protein.

## Nub-PD is essential for mitosis and nuclear divisions during early *Drosophila* embryogenesis

To examine the mitotic functions of the Nub-PD isoform *in vivo* during development, we utilized the syncytial blastoderm stage of *Drosophila* embryos, which entirely relies on maternally provided transcripts due to the absence of transcription (Fig. S2A). Early *Drosophila* embryogenesis is characterized by rapid and synchronous nuclear divisions switching between S and M phases, creating a syncytial blastoderm (Fig. S2A) (Harrison et al., 2023). Previous studies have shown *nub* mRNA expression in syncytial embryos (Lécuyer et al., 2007; Wilk et al., 2016). Consistent with this, robust expression of *nub-RD*, but not of *nub-RB*, was observed in 0–2-h-old embryos (Fig. S2B). Unfertilized oocytes dissected from female ovaries revealed no Nub protein immunostaining, indicating that *nub* mRNA is not translated during oogenesis (Fig. S2C). However, Nub immunostaining was evident throughout newly laid embryos at nuclear division cycle 1 (NC1) (Fig. S2D), and in nuclei at NC7 and NC12 before zygotic genome activation (ZGA) (Fig. S2F), demonstrating efficient translation of *nub* mRNA during the syncytial stages. Furthermore, embryos laid by females mated with 'spermless'

*tudor* (*tud*) males also expressed Nub protein (Fig. S2E), reinforcing that *nub is* expressed from the female germline and prior to ZGA. It has been previously reported that a large fraction of maternally provided transcripts are translated during ovulation in a process called egg activation (Harrison et al., 2023; Krauchunas and Wolfner, 2013). Our results demonstrate that *nub* is one of those genes that is maternally transcribed and subsequently translated during egg activation. Importantly, we observed strong Nub immunostaining around the mitotic spindle of the first nuclear division (Fig. S2D–F, arrowheads) suggesting that Nub-PD is translated very early and is recruited to dividing nuclei from the first embryonic mitoses.

Next, we investigated the mitotic progression during early embryogenesis in embryos with reduced Nub-PD levels using a recessive hypomorphic mutation, *nub¹*, with strongly diminished *nub-RD* expression due to a transposon located just upstream of the *nub-RD* transcription start site (Fig. S2G–I″) (Ng et al., 1995). Despite a previous report showing diminished expression of the *pdm2* paralog in *nub¹* mutant wing discs (Loker and Mann, 2022), we found no effect on Pdm2 protein expression in *nub¹* syncytial embryos (Fig. S2J). Homozygous *nub¹* females are fertile, but the embryo hatch rate was only 60% (Fig. S2K), demonstrating compromised development. This enabled us, however, to analyze mitosis in early syncytial embryos with reduced Nub-PD levels. By following nuclear divisions in *nub¹* embryos, we noticed disrupted nuclear distribution in the cortex with areas devoid of nuclei, especially at NC10-11 (Fig. 2A,B), and reduced nuclear occupancy at NC10–NC13 compared to control embryos (Fig. 2C). Furthermore, in both *nub¹* mutants and *MTD>nub RNAi* embryos, we observed normal chromosome alignment at metaphase (Fig. 2D,E) but obvious chromosome mis-segregation during anaphase, including abnormal anaphase chromosome geometry. These defects persisted into telophase as unresolved chromosome bridges (Fig. 2F–I), leading to nuclear fallout (NUF), a well-described endpoint for defective nuclei (Rothwell et al., 1998; Takada et al., 2003). Although partial recovery in nuclear occupancy was observed in some *nub¹* embryos at NC12–13 (Fig. 2C), we conclude that the pronounced division failure is likely a major cause of the poor embryo survival and low hatch rate of *nub¹* mutants (Fig. S2K).

To further explore the role of Nub-PD in nuclear divisions we performed time-lapse microscopy during the last four syncytial mitotic cycles (NC10–13) in wild-type (*wt*) and *nub¹* embryos expressing the RFP-tagged histone H2A variant His2Av (His2A–RFP) (Pandey et al., 2005). First, we investigated whether the defective nuclear divisions were the result of aberrant DNA synthesis (S-phase) or mitosis (M-phase). During syncytial nuclear divisions, nuclei alternate between S and M phases, with no gap phases (G-phases) (Glover, 1991; Harrison et al., 2023) (Fig. S2A). The duration of the S phase (Fasulo et al., 2012) was indistinguishable between *nub¹* and control embryos (Fig. 3A,B), but the M phase was significantly longer in *nub¹* mutant embryos in NC11–13 (Fig. 3A,C). Additionally, time-lapse imaging revealed five mitotic defects in *nub¹* embryos – clustered or aggregated nuclei, asymmetrically spread nuclei, asynchronous divisions, arrested divisions, and cortical gaps with NUF (Fig. 3D–F; Fig. S3A–C and Movies 1–3). Approximately 55% of *nub¹* embryos displayed clustered or aggregated nuclei, and 45% exhibited large NUF patches, consistent with the phenotypic analysis based on immuno-staining of fixed embryos. Only 12–16% of wild-type controls showed such nuclear division phenotypes. Moreover, *nub¹* embryos showed significant asynchrony and failed to maintain spatial nuclear arrangements, leading to asymmetric spreading of nuclei in 25% of embryos (Fig. S3A–C). Small subsets of *nub¹* embryos (15%),

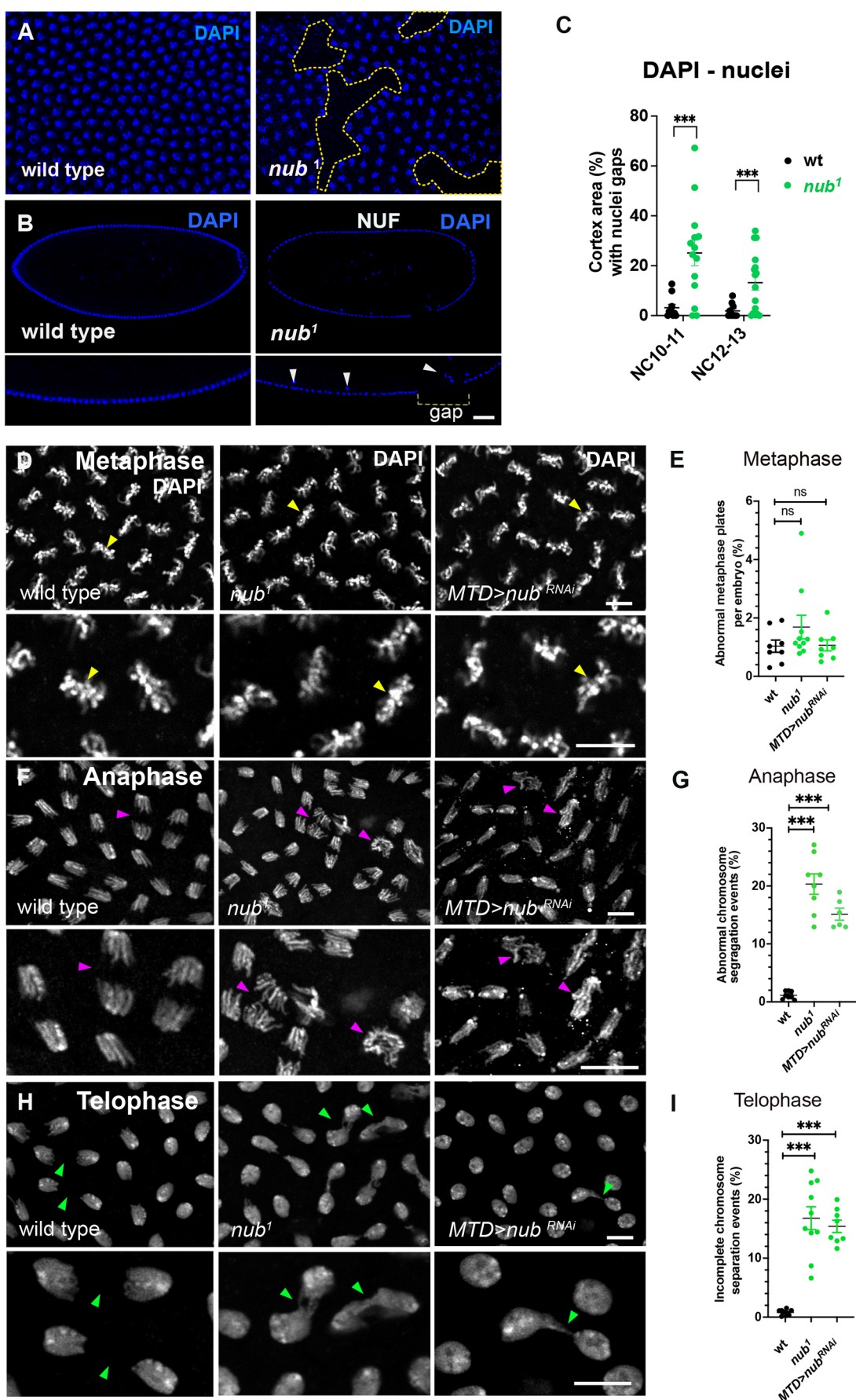

**Fig. 2.** See next page for legend.

**Fig. 2. Nub is required for efficient nuclear division and faithful chromosome segregation in *Drosophila* syncytial embryos.** (A) Confocal images displaying the cortex of wild-type and *nub¹* embryos stained with the nuclear probe DAPI. Dashed lines, nuclei-free cortical areas. (B) Confocal middle sections of wild-type and *nub¹* embryos stained for DAPI. Images below are zoomed-in areas of the cortex, shown in the upper images. Arrowheads depict nuclei falling into the inner yolk region. (C) Scatter plot showing the percentage of the cortical area with nuclei gaps in wild-type (*n*=18) and *nub¹* (*n*=27) mutant embryos during indicated nuclear cycles (NC). \*\*\*$P<0.001$ (unpaired two-tailed *t*-tests). (D,F,H) Representative confocal images of *Drosophila* pre-blastoderm embryos at metaphase (D), anaphase (F) and telophase (H) stained with DAPI to visualize chromosomes. Wild-type embryos show properly aligned metaphase chromosomes (yellow arrowheads), normal segregation during anaphase (purple arrowheads), and evenly sized daughter nuclei in telophase (green arrowheads). In contrast, *nub¹* mutant and *MTD>nub RNAi* embryos display chromosome misalignment, lagging chromosomes, and unequal chromosome distribution, indicating disrupted mitotic progression. (E,G,I) Quantification of abnormal chromosome configurations in metaphase (E), anaphase (G) and telophase (I). The percentage of nuclei showing misaligned chromosomes (metaphase), abnormal chromosome segregation (anaphase), or incomplete or unequal segregation (telophase) was significantly increased in *nub¹* and *MTD>nub RNAi* embryos relative to that in wild type. Each data point represents one embryo; mean±s.e.m. shown. \*\*\*$P<0.001$; ns, not significant (unpaired two-tailed *t*-test). Scale bars: 10 μm.

compared to wild type (3%), were arrested and failed to progress with subsequent nuclear divisions (Fig. S3B).

## Loss of Nub-PD affects spindle organization and anaphase elongation

To understand the primary cause of the phenotypes of dividing nuclei upon Nub-PD protein loss, we examined the mitotic spindle organization by co-immunostaining for β- and γ-tubulin in wild-type and *nub¹* embryos. As shown in Fig. 4A,B and Movies 4,5, *nub¹* embryos displayed disorganized spindles with detached microtubule bundles, particularly during the NC11–13 of syncytial development. We also frequently observed free centrosomes at the cortex, and spindles disconnected from centrosomes below the cortex, which then fell into the interior of the embryo for spindle microtubule de-polymerization (Fig. 4C).

Live-imaging of embryos expressing the microtubule plus-end binding protein 1 (EB1) tagged with GFP (EB1–GFP) (Piehl et al., 2004) showed that there was normal spindle dynamics in control embryos (Fig. 4D), as previously reported (Karr and Alberts, 1986; Kellogg et al., 1988; Sullivan and Theurkauf, 1995). In contrast, *nub¹* embryos produced mitotic spindles during metaphase, but these failed to maintain their proper size and shape (Fig. 4D–F). Pre-anaphase B spindle length (pole-to-pole distance) was shorter in *nub¹* embryos compared to in control embryos (Fig. 4G) at NC9. Spindles progressively collapsed and detached from centrosomes in metaphase and anaphase, leading to accumulation of free centrosomes at the cortex (Fig. 4D,E), consistent with the γ-tubulin staining (Fig. 4B,C). The centrosome density (centrosomes per μm²) was unchanged in *nub¹* mutants compared with wild type, arguing against defects in centriole duplication or centrosome amplification (Fig. 4H).

Tracking individual spindles in EB1–GFP embryos revealed significant delays in the metaphase-to-anaphase transition in *nub¹* embryos at both NC10 and NC11 (Fig. 4I,J), consistent with the defects observed at NC9. Wild-type spindles showed dynamic changes in the end of pre-anaphase B with an average elongation rate of 0.047 μm/s. In contrast, *nub¹* mutant spindles elongated more slowly, averaging 0.027 μm/s (Fig. 4K). These phenotypes suggest a role of Nub-PD in anaphase spindle elongation. Taken together, our

data indicate that Nub-PD plays a pivotal role in spindle organization and elongation to sustain rapid syncytial nuclear divisions.

We then aimed to restore the mitotic function in *nub¹* genetic background by introducing a functional *nub* gene copy via a *nub* locus duplication on the third chromosome (CH321-40A01). This duplication line showed increased Nub immunostaining compared to wild type in zygotic embryos (Fig. S4A). The *nub* locus duplication provided a significant rescue of the NUF phenotype (Fig. S4B,C), and of the mitotic spindle organization in *nub¹* mutants (Fig. S4D,E). These results confirm that the mitotic failures were specifically due to the loss of *nub*, and highlight the essential contribution of Nub-PD for spindle stability, and consequently for proper chromosome segregation during mitosis.

## A critical and non-transcriptional role of Nub-PD in early embryos

To further validate the essential role of Nub-PD protein in the early embryo we specifically depleted Nub-PD protein without affecting the maternally loaded *nub-RD* mRNA, using targeted degradation of GFP fusion proteins (deGradFP) (Caussinus et al., 2011) (Fig. 5A). We first created Nub-PD and Nub-PB proteins endogenously tagged with the superfolder GFP (sfGFP), and then generated two independent transgenic fly lines (*sfGFPnub-PB* and *sfGFPnub-PD*) by CRISPR/Cas9-mediated homology-directed repair (HDR) (Fig. 5B). Both sfGFP-tagged transgenic lines were homozygous viable and fertile. Furthermore, the reduced nuclear-to-cytoplasmic (N/C) ratio at NC11–12 in *nub¹* mutants was returned back to wild-type levels in heterozygous *sfGFP-nub-PD/nub¹* embryos generated by introducing the *sfGFP-nubPD* allele into the *nub¹* background (Fig. S5A). The adult life span of *sfGFP-nubPD* and *sfGFP-nub-PB* was near wild-type levels (Fig. S5B), demonstrating that the tagged protein forms are functionally competent *in vivo*. Consistent with the mRNA measurements (Fig. S2B), early *sfGFP-nubPD* embryos, but not *sfGFP-nubPB* embryos, showed a GFP signal (Fig. 5C,D).

Next, the *sfGFPnub-PD* line was crossed with the previously described *hunchback*-deGradFP (*hb*-deGradFP) and *nanos*>deGradFP (*nos*>deGradFP) transgenic lines (Gaskill et al., 2021; Vazquez-Pianzola et al., 2022). The *hb*-minimal maternal promoter and the *nos* promoter drive the expression of *deGradFP* mRNA during oogenesis, which subsequently is translated into active deGradFP protein upon egg activation (ovulation), leading to degradation of newly translated sfGFP-tagged proteins in early embryogenesis. The sfGFP–NubPD fluorescence was strongly reduced by expression of the *deGradFP* in early syncytial embryos, confirming prominent degradation of GFP-tagged Nub-PD protein (Fig. 5E; Fig. S5C). These Nub-PD-depleted embryos exhibited asynchronous nuclear divisions, chromosome segregation defects and cortical areas with nuclei gaps (Fig. 5F–F″; Fig. S5C–C‴). Additionally, degradation of sfGFP–NubPD caused severe spindle organization phenotypes, characterized by detached microtubule bundles similar to those observed in *nub¹* mutants (Fig. 5G). This confirms that loss of Nub-PD in transcriptionally silent syncytial embryos, but not during oogenesis when transcription is functionally robust, causes the aberrant mitotic phenotypes.

To further confirm the non-transcriptional function of Nub in nuclear divisions, we acutely inhibited Nub protein in wild-type embryos by microinjecting an anti-Nub antibody during early syncytial nuclear cycles. Live imaging of wild-type embryos expressing His2A–RFP, micro-injected at NC8 with Nub antibody, recapitulated the effects observed in *nub¹* embryos (Fig. 5H,I; Fig. S5D). Antibody injection led to a 75% incidence of clustered or aggregated nuclei, compared to 15% in controls, causing chromosome segregation phenotypes (Fig. 5J; Movie 6). These

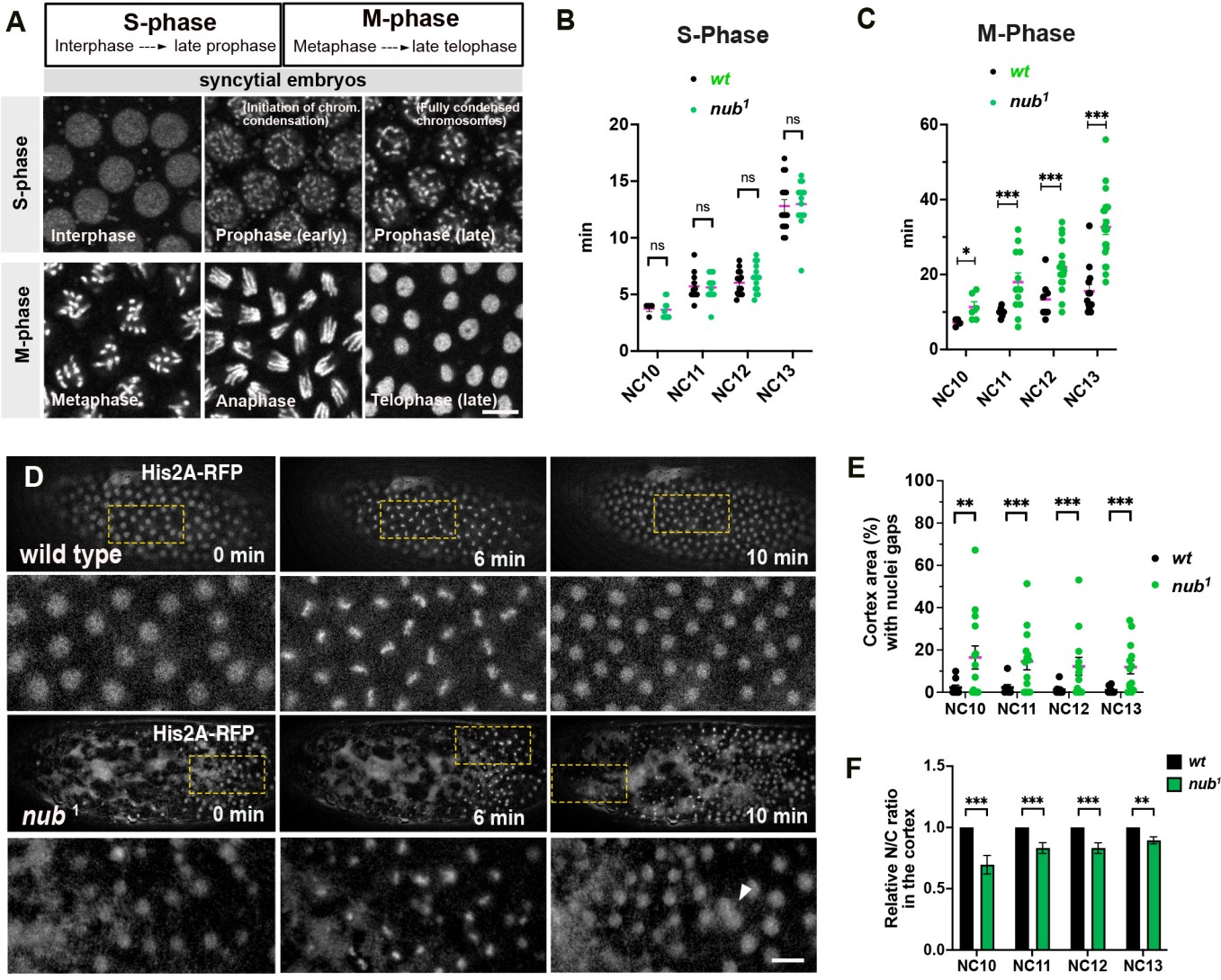

**Fig. 3. Loss of Nub disrupts the timing and organization of nuclear divisions during early syncytial development.** (A) Reference panel from live imaging recordings of dividing nuclei in *Drosophila* syncytial embryos expressing the His2A–RFP reporter, showing how S-phase (interphase to late prophase) and M-phase (metaphase to late telophase) were defined during nuclear cycles (NC10–13). The phases were classified according to Fasulo et al. (2012) and Yu et al. (2000). Scale bar: 10 μm. (B,C) Scatter plots showing the time in minutes (min) spent in S phase (B) and M phase (C) in wild type (*n*=48) and *nub¹* (*n*=85) embryos during NC10–13. *P<0.05; ***P<0.001; ns, not significant (unpaired two-tailed *t*-tests). (D) Fluorescent images from time-lapse recordings showing live wild-type and *nub¹* mutant embryos at NC10–11. Embryos are expressing the His2A–RFP reporter. Images below depict magnified areas of the embryo cortex indicated by the rectangular frames in the upper images. White arrowhead indicates attached and mitotically arrested nuclei. Scale bar: 10 μm. (E,F) Plots showing the percentage of cortical area with nuclei gaps (E), or the relative nuclear to cytoplasmic ratio (N/C ratio) (F), per embryo in wild-type (*n*=47) and *nub¹* (*n*=57) mutant embryos during different nuclear cycles (NC10–13). Values were normalized to wild type. **P<0.01; ***P<0.001 (unpaired two tailed *t*-tests). All error bars show mean±s.e.m.

phenotypes were particularly strong near to the injection site, marked by labeled dextran (3 kDa). As expected, the nuclei from these highly irregular divisions subsequently fell into the interior of the embryo (NUF phenotype), creating large cortical nuclei gaps (Fig. 5I; Fig. S5D). Taken together, our data demonstrate that Nub-PD is required for an essential non-transcriptional function in mitotic progression and nuclear division *in vivo*.

In *nub¹* embryos, mitotic defects were prominent at NC10–13 but less evident at earlier NCs (Fig. S5F), despite the finding that Nub localized to the mitotic spindle during the first nuclear division (Fig. S2F). We hypothesized that the less pronounced phenotype at early NCs reflects residual Nub-PD protein in the hypomorphic *nub¹* allele (Fig. S2I–I″). To investigate this, we combined the *nub¹* mutant with *nub* RNAi knockdown using the maternal

driver matα4-Gal4-VP16. This led to more severe mitotic defects, including chromosome segregation errors (Fig. S5E–G) and spindle organization defects (Fig. S5H–I) as early as NC5, supporting the hypomorphic nature of the *nub¹* allele. Given that the earliest signs of zygotic gene transcription have been reported at NC8 (Edgar and Schubiger, 1986; Erickson and Cline, 1993; Kwasnieski et al., 2019; Pritchard and Schubiger, 1996), the mitotic defects recorded here at NC5 further argues for a non-transcriptional role of Nub-PD during syncytial divisions.

### Dynamic subcellular localization of Nub-PD protein during mitosis

To further elucidate the role of Nub-PD during mitosis, we followed its localization during the mitotic phases in *Drosophila* syncytial

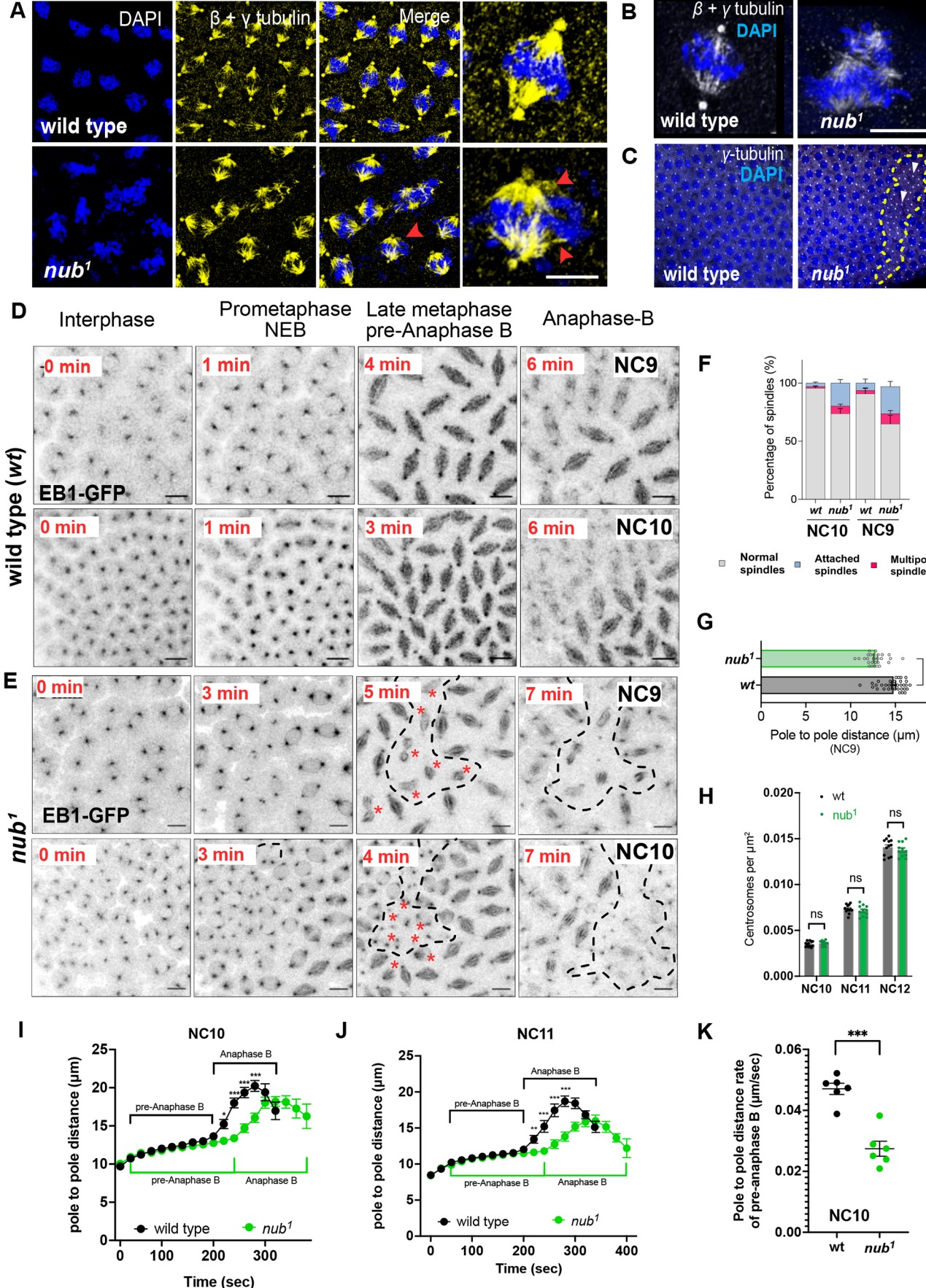

**Fig. 4.** See next page for legend.

**Fig. 4. Nub-PD is necessary for proper spindle organization and elongation in syncytial-blastoderm stage embryos.** (A) Representative images showing $w^{1118}$ and $nub^1$ syncytial embryos stained for β-tubulin, γ-tubulin (yellow) and DNA with DAPI (blue). Disorganized $nub^1$ spindles with detached microtubular bundles (red arrowheads) are shown (with magnified images to the right). (B) Confocal images displaying single spindles of wild-type and $nub^1$ mutant embryos stained for β-tubulin (spindles, white) and γ-tubulin (centrosomes, white), and DNA with DAPI (blue). (C) Confocal images of the cortex of wild-type and $nub^1$ mutant embryos stained for γ-tubulin (white) and DNA with DAPI (blue). Cortical areas with centrosomes and no nuclei are indicated (dash lines). Arrowheads denote free centrosomes. (D,E) Representative time-lapse images of wild-type (D) and $nub^1$ mutant (E) embryos expressing EB1–GFP in NC9 and NC10. The corresponding time (min) from interphase to anaphase is indicated. Dotted areas, along with asterisks indicate groups of disorganized or attached spindles and free centrosomes, respectively. (F) The mean±s.e.m. percentage of normal, multipolar and attached spindles is shown for $wt$ and $nub^1$ mutant embryos at NC9 and NC10. $n$=30–50 mitotic spindles, $N$=5 embryos. Multipolar and attached spindles in $nub^1$ mutant embryos were significantly ($P<0.01$) different from $wt$ as assessed by two-way ANOVA (multiple comparison analysis). (G) Spindle length (pole-to-pole distance) in $nub^1$ and wild-type embryos at NC9. Mean±s.d. ($wt$, $n$=35, $nub^1$, $n$=27). ***$P<0.001$ (unpaired two-tailed $t$-test). (H) Quantification of centrosome density per μm$^2$ at nuclear cycles NC10–12 showing no significant difference between wild-type and $nub^1$ embryos. Mean±s.e.m. ($wt$, $n$=12–14; $nub^1$, $n$=8–13 embryos). ns, not significant (unpaired two-tailed $t$-test). (I,J) Pole-to-pole distance (μm) as a fraction of time (s) in wild-type and $nub^1$ dividing spindles expressing EB1–GFP during metaphase, pre-anaphase B and anaphase B at NC10 (I) and NC11 (J). Each time point represents the average data from multiple spindles ($n$) and embryos ($N$). Mean±s.e.m.; $n$=128–137, $N$=9–10 embryos per genotype. *$P<0.05$, **$P<0.01$; ***$P<0.001$ (logistic regression model using the Chi-squared test). (K) Graph showing mean±s.e.m. rates of spindle pole-to-pole distance (μm/s) in $wt$ ($N$=6 embryos, $n$=78 spindles) and $nub^1$ ($N$=6 embryos, $n$=67 spindles) mutant embryos expressing EB1–GFP. ***$P<0.001$ (unpaired two tailed $t$-test). Scale bars: 10 μm.

embryos. Nub immunostaining was very dynamic, with punctate staining in interphase nuclei (Fig. 6A). After nuclear envelope breakdown (NEB), at the onset of metaphase, Nub staining was restricted within the spindle bundles, being more intense around the chromosomes, but not on centrosomes (Fig. 6A). At anaphase–telophase, Nub was recruited back to the newly formed nuclei and was cleared from the central spindle or midzone (Fig. 6A,C), indicating its active translocation during mitosis. To rule out fixation artifacts (Teves et al., 2016), we fixed embryos at low formaldehyde concentration (0.5%) and observed the same Nub enrichment within the spindle envelope (Schweizer et al., 2014) (Fig. S6A). Furthermore, live imaging of S2 cells transfected with constructs expressing mKO2- and mRFP-tagged Nub-PD also showed similar spindle enrichment during metaphase (Fig. 1H; Fig. S1J). Analyses of endogenously GFP-tagged Nub-PD in two independent knock-in fly lines (*sfGFP-nubPD*) and *PBac (nub-GFP)* using anti-GFP antibodies revealed identical localization pattern (Fig. 6B; Fig. S6B). In contrast, staining with an antibody against the related paralog Pdm2 (Fig. S6D,E), showed a different localization pattern. Although Pdm2 was present in interphase nuclei like Nub, it was absent from spindle microtubules during metaphase, but enriched in the midzone and centrosomes (Fig. S6C). These findings clearly demonstrate a dynamic and specific localization of Nub-PD during mitosis (Fig. 6C), consistent with its role in spindle organization and morphology in both syncytial embryos and S2 cells.

## Nub-PD localization requires intact spindle microtubules in an interdependent manner

To determine whether the Nub-PD enrichment at metaphase spindles depends on intact microtubules, we undertook a non-invasive

cold-induced de-polymerization of microtubules at the syncytial blastoderm stage (Hayward et al., 2014) (Fig. 6D,E). Incubation of wild-type embryos at 2°C (60 min) completely eliminated tubulin staining at metaphase and also abolished Nub-PD enrichment around the spindles. Spindles quickly reformed upon recovery at room temperature, but Nub-PD did not reappear around the spindles until after ~10 min, presumably at the following nuclear division (Fig. 6F,G). This indicates that Nub-PD localization relies on intact spindle microtubules and possibly other mitotic factors that support its spindle targeting.

The nuclear proteins Skeletor, Megator, East and Chromator are localized throughout spindle microtubules by forming a matrix during the partially open mitosis of *Drosophila* syncytial embryos (Schweizer et al., 2014; Yao et al., 2018). This spindle matrix has been proposed to act as a mechanical element that mediates the interaction and movement of the kinetochore towards the poles (Qi et al., 2004; Zheng, 2010). To analyze a possible connection between Nub-PD and this matrix in maintaining spindle integrity, we carried out immunostaining of the spindle matrix protein Megator. Megator showed a similar localization pattern to Nub-PD in wild-type embryos, and this did not change in $nub^1$ mutant embryos (Fig. S6F). In contrast to Nub-PD, Megator remained intact in a mesh-like structure around the spindles after cold treatment (Fig. S6G), consistent with other studies using nocodazole as depolymerizing factor (Yao et al., 2018). Thus, Nub-PD but not Megator requires the presence of spindle microtubules for its localization during mitosis (Fig. 6F,G; Fig. S6G), indicating distinct localization requirements. In conclusion, our data argue against Nub-PD being an interacting partner within the matrix. Rather, our findings indicate that Nub-PD associates with microtubules to maintain spindle shape, length and integrity during the rapid syncytial nuclear divisions.

Next, we used the same assay to examine how efficient the $nub^1$ nuclei are in re-nucleating and re-stabilizing microtubules once conditions become permissive again. We subjected wild-type and $nub^1$ embryos to cold treatment and followed the microtubular regrowth at room temperature, and documented it live or after fixation. In untreated embryos, microtubule spindle regrowth was visible within 2 min (Fig. 7A–C; Fig. S6H,I). On the contrary, cold-treated $nub^1$ embryos showed delayed microtubule regrowth during metaphase (Fig. 7A–C), and live imaging revealed that proper spindle architecture was not maintained at anaphase onset (Fig. 7D). Instead, abnormally elongated astral-like microtubules were formed, extending from the centrosomes (Fig. 7E). Notably, the loss of Nub-PD disrupted the organization of the interpolar microtubule (iMTs) array in the midzone, leading to unstable spindles (Fig. 7D). These results further support the role of Nub-PD in maintaining microtubule stability in spindles, enabling them to resist poleward forces during anaphase. Additionally, a portion of $nub^1$ cold-treated embryos (30%) failed to reform spindle microtubules (Fig. S6H,I), suggesting a possible contribution of Nub-PD for efficient spindle microtubule re-nucleation after induced de-polymerization. Taken together, our data argue that Nub-PD is important for both the stability and reformation of the mitotic spindle, with implications for microtubule nucleation and organization during mitosis.

Next, we investigated whether human POU2F1 plays a similar role in spindle reconstitution and regrowth. Using the same cold treatment assay (for 30 min), we found a defect in spindle reassembly upon POU2F1 depletion. In control cells, bipolar spindles rapidly reformed within a few minutes after temperature recovery, whereas POU2F1-depleted cells showed delayed and disorganized spindle regrowth (Fig. 7F). Quantification confirmed a significantly reduced spindle reformation rate and lower spindle

Journal of Cell Science

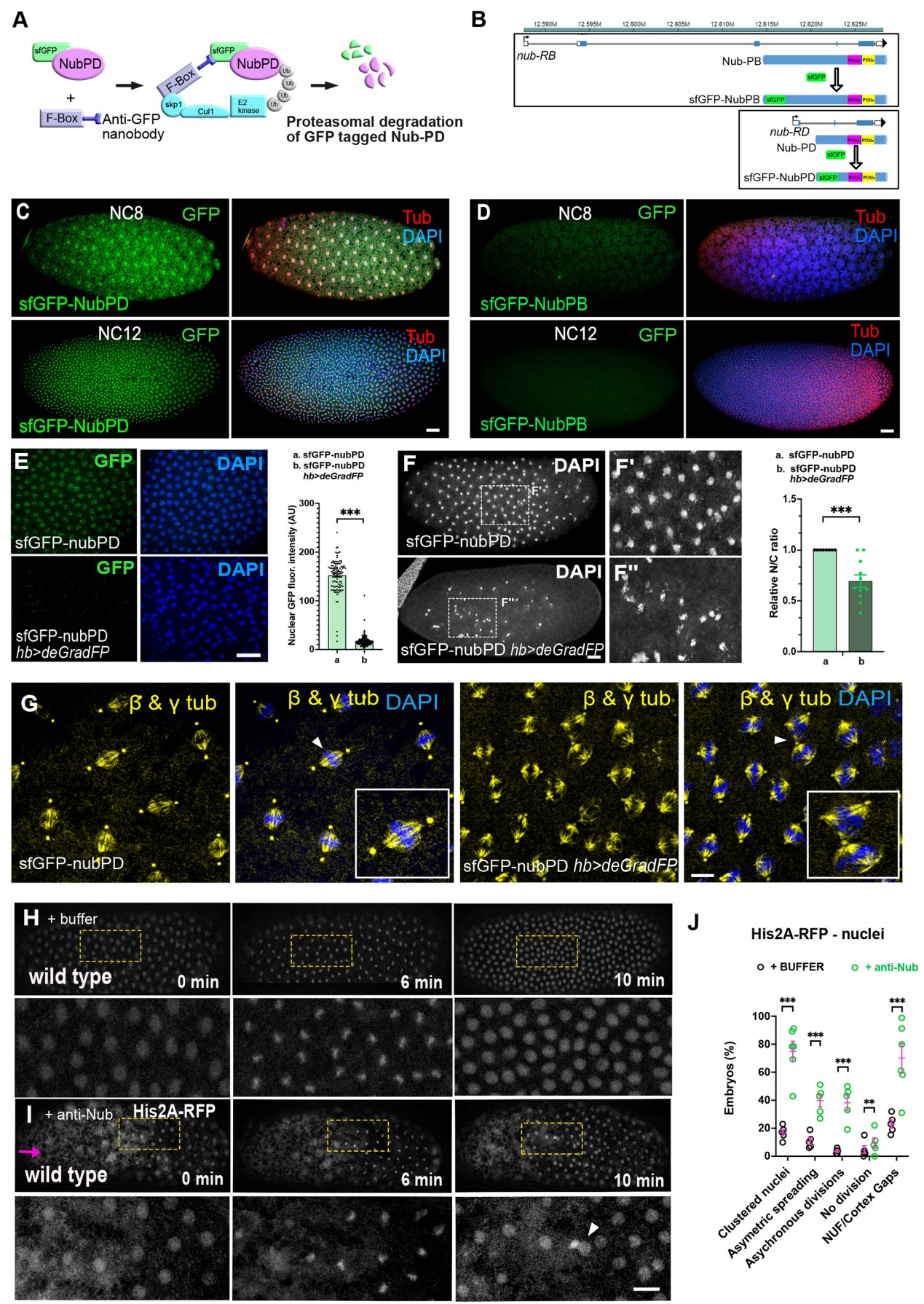

**Fig. 5.** See next page for legend.

**Fig. 5. Impairment of Nub-PD in syncytial embryos demonstrates a non-transcriptional role in mitosis.** (A) Schematic representation of the deGradFP (utilizing a ubiquitin-proteasome pathway) for degradation of GFP-tagged Nub-PD protein in embryos. (B) Schematic representation of sfGFP-tagged Nub protein isoforms, sfGFP–NubPB and sfGFP–NubPD. Independent fly lines were generated by CRISPR-Cas9-mediated genome editing, tagging endogenous Nub-PB and Nub-PD proteins with sfGFP. (C) Fluorescence images of *sfGFP-NubPD* (green) in embryos at NC8 and NC12. β-tubulin (red) and DAPI (blue). (D) Fluorescence images of *sfGFP-NubPB* (green) in blastoderm-staged embryos at NC8 and NC12. β-tubulin (red) and DAPI (blue). Images in C and D representative of three experimental repeats. (E) Representative confocal images of control *sfGFP-Nub-PD* embryos (top) and embryos expressing *hb>deGradFP* (bottom), stained with DAPI and antibodies against GFP. Quantification (right) shows a strong reduction in nuclear GFP intensity (arbitrary units, A.U.) upon *deGradFP* expression. Mean±s.e.m. (*wt*, *n*=106; *nub¹*, *n*=160 nuclei). ***P<0.001 (unpaired two tailed *t*-test). (F) Fluorescence images of control (*sfGFP-NubPD*) and *sfGFP-NubPD* embryos during NC8, stained for DNA by DAPI (white). F′ and F″ show magnified views of areas marked. Quantification (right) reveals a significant reduction in nuclear to cytoplasmic (N/C) ratio upon *sfGFP-NubPD* depletion. Mean±s.e.m. (*wt*, *n*=9; *nub¹*, *n*=11 embryos). ***P<0.001 (unpaired two-tailed *t*-test). (G) Confocal images showing the organization of the mitotic spindles in control (*sfGFP-NubPD*) and *sfGFP-NubPD*; *hb>deGradFP* embryos. Tubulin (β-tubulin+γ-tubulin, yellow) and DAPI (blue). Magnified views of areas marked are shown in the corresponding images. Images representative of three experimental repeats. (H) Fluorescence images from time-lapse recordings showing live wild-type buffer-injected embryo at NC10–11. Embryos are expressing the His2A–RFP reporter. Images below depict magnified views of areas of the embryo cortex indicated by the rectangular frames in the upper images. (I) Fluorescence images from time-lapse recordings showing live wild-type embryos injected with anti-Nub antibody. Embryos are expressing the His2A-RFP reporter. Images below indicate magnified views of areas of the embryo cortex indicated by the rectangular frames in the upper images. The injection site is indicated by an arrow (magenta). Arrowhead indicates abnormal nuclei division. (J) Graph showing the percentage of wild-type embryos injected with buffer (*n*=28) and anti-Nub (*n*=37) antibody with attached, asymmetric, asynchronous, arrested nuclei and nuclear fall-out (NUF) phenotypes. Values represent mean±s.e.m. of five repeated experiments (*N*=5). **P<0.01; ***P<0.001 (unpaired two tailed *t*-test). Scale bars: 30 μm (C–F); 10 μm (G–I).

coherency in POU2F1-knockdown cells compared to controls (Fig. 7G,H). These results demonstrate that POU2F1 is required for efficient spindle microtubule regrowth following de-polymerization. Together, these findings suggest that Nub-PD and POU2F1 might share the same functions in maintaining stable spindle microtubule dynamics during mitosis, when microtubules are continuously reorganized (Fig. 7I,J).

### Factors affecting Nub-PD recruitment to mitotic spindles

To identify the molecular interactions controlling the dynamic recruitment of Nub-PD during nuclear division cycles, we carried out a targeted maternal RNAi-knockdown screen of components of the mitotic spindle machinery during syncytial nuclear divisions, followed by Nub immunostaining. We analyzed 29 genes, using 45 independent RNAi or mutant fly lines, and out of these, nine genes affected the localization of Nub during mitosis (Table S1). Several of these genes specifically altered the distribution of Nub-PD during M-phase but not during S-phase (interphase) (Fig. S7B,C) and this pattern was consistent between early and late NCs (Table S1). We consider the effects of these latter genes to be more specific and likely to reflect a direct role in regulation of Nub-PD localization, rather than secondary consequences of general cell cycle disruption. More specifically, downregulation of the kinases Nek2 and Niki drastically reduced Nub-PD localization specifically in metaphase, suggesting phosphorylation by these kinases is essential for the

translocation of Nub-PD to the spindles (Fig. S7A–C). Similarly, knockdown of Klp61F (the ortholog of vertebrate kinesin-5 or EG5) and Klp3A abrogated Nub-PD localization in metaphase but not in interphase (Fig. 8D; Fig. S7A–C). The expression level of Klp61F in syncytial mitosis is controlled by the Crb, Galla-2 and Xpd (CGX) complex (Hwang et al., 2020). Our analysis showed that RNAi of Crb, but not of Galla-2 or Xpd, affected Nub-PD localization around the spindles (Fig. S6A; Table S1). Crb and Klp61F are localized in spindles, and their knockdown or mutation causes mitotic phenotypes similar to those seen in *nub¹* mutants during syncytial divisions (Hwang et al., 2020). The CPC is a highly conserved master regulator of mitosis that is required for spindle and kinetochore assembly. In mammalian cells, it is composed of four subunits: the enzymatic component Aurora B (AurB) and the three regulatory and targeting components Incenp, survivin and borealin (Carmena et al., 2012). Our analysis shows that knockdown of AurB, Incenp and of the borealin-related protein Deterin (also known as Survivin) abolished Nub-PD staining in interphase and prophase nuclei and metaphase spindles (Fig. 8A–C; Fig. S7A–C, Table S1). Furthermore, Cdk1 disruption affected Nub-PD localization both in interphase and metaphase, indicating that Nub-PD localization, and presumably its function, is downstream of the mitotic-entry signals (Table S1). Because knockdowns of AurB, Incenp and Deterin caused strong mitotic and developmental defects (Table S1), we cannot rule out that the severe mis-localization of Nub in these embryos (interphase and metaphase) is a secondary effect due to the overall mitotic or developmental phenotypes caused by depletion of these mitotic components. However, downregulation of Crb, Klp3A, Klp61F, Nek2 and Niki caused reduced Nub-PD signal within metaphase spindles, but not in interphase, and overall milder spindle mitotic defects under the RNAi conditions. Importantly, we performed these RNAi experiments under moderate depletion conditions, chosen to preserve overall mitotic progression and to enable assessment of Nub-PD localization. We acknowledge that this strategy might have reduced the sensitivity to detect additional localization phenotypes of Nub-PD compared to complete knockdown. This is consistent with our observations in *nub¹ᐟ⁺* heterozygous embryos (Fig. S5A), where Nub-PD signal is reduced but mitosis proceeds normally. Taken together, our work identifies a regulated and precisely timed translocation of Nub-PD from the chromatin to the spindle envelope during the rapid nuclear cycles in the syncytial embryo.

### DISCUSSION

Nub has been shown to be required for proliferation in the *Drosophila* embryonic nervous system, wing discs and midgut epithelium (Cifuentes and García-Bellido, 1997; Doe, 2017; Ng et al., 1995; Tang et al., 2018; Tran and Doe, 2008; Tsuji et al., 2008). However, how Nub controlled proliferation during development was not known and its role in mitosis had not been explored. Several transcription and splicing factors have been suggested to confer direct mitotic functions based on their localization to the mitotic apparatus, but very few have been confirmed experimentally by loss-of-function analyses (Pellacani et al., 2018; Somma et al., 2020). Here, we show that the Nub-PD isoform is not just a transcriptional regulator in interphase but also a crucial mitotic factor during early embryogenesis. We demonstrated that it is required for timely progression of the rapid and synchronous syncytial nuclear divisions through maintaining mitotic spindle integrity and elongation. Importantly, this function is independent of its transcriptional activity, revealing a previously unrecognized, non-transcriptional role of Nub-PD in early development.

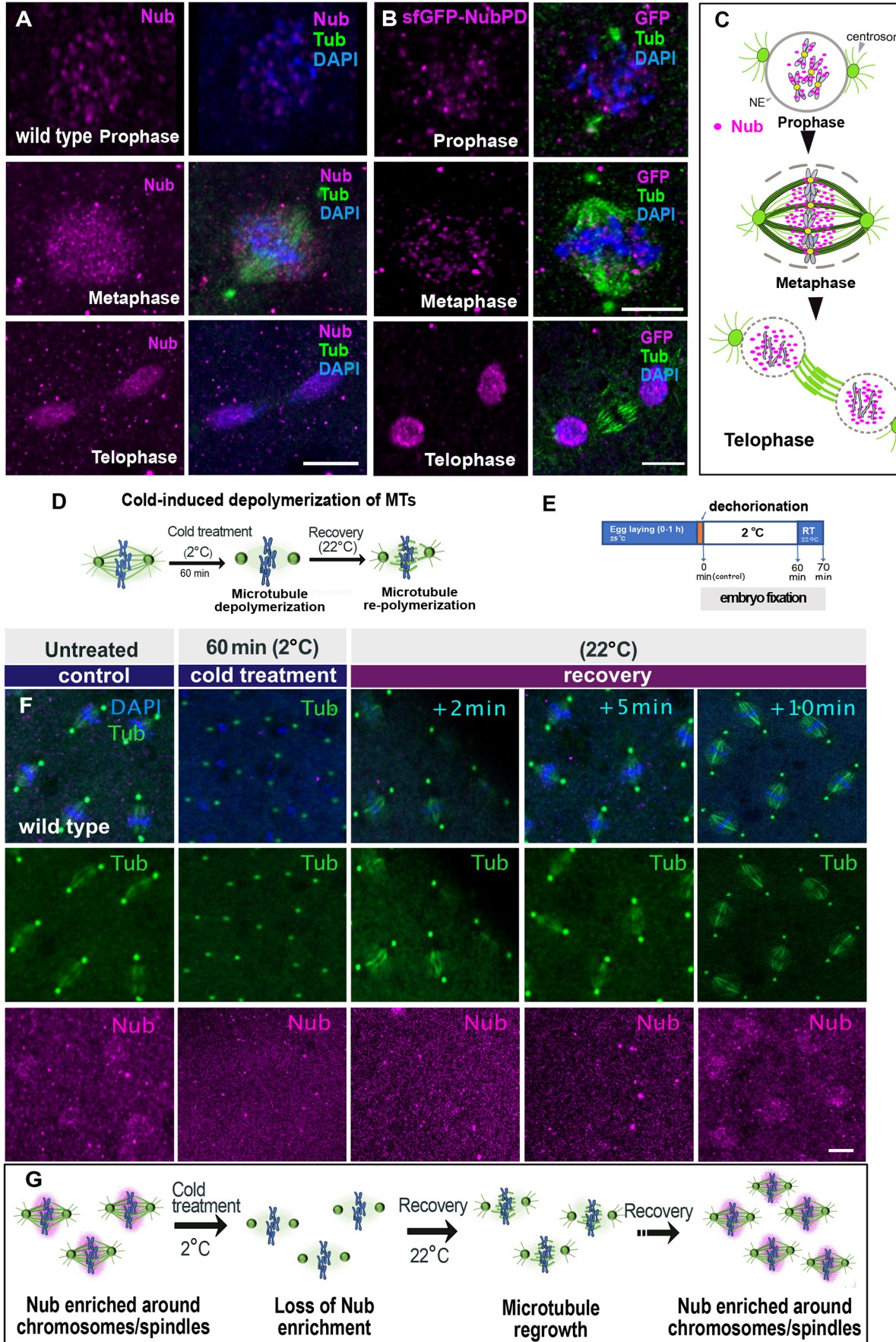

**Fig. 6.** See next page for legend.

**Fig. 6. The dynamic localization of Nub within the mitotic spindle requires intact spindle microtubules.** (A) Airy-scan confocal images (projections) of wild-type syncytial embryos showing the localization of Nub-PD (magenta), α- and γ-tubulin (Tub, green), and DNA (blue) during prophase, metaphase and telophase. (B) Airy-scan confocal projections showing *sfGFP-NubPD* syncytial embryos stained with antibodies for GFP (magenta), α- and γ-tubulin (Tub, green), and DNA with DAPI (blue) during prophase, metaphase,and telophase. (C) Graphical illustration describing the dynamic localization of Nub-PD (magenta) during prophase, metaphase and telophase. Centrosomes (green); nuclear envelope (NE, gray). (D) Schematic illustration of the spindle effects by cold-induced treatment during metaphase. Microtubules (MTs) depolymerized and spindles disassembled upon cold treatment, but centrosomes remained intact. (E) Diagram showing the procedure of cold-induced treatment in syncytial embryos. Embryos were shifted to room temperature up to 10 min to allow microtubule regrowth before fixation and staining. (F) Confocal images showing representative spindles of wild-type syncytial embryos showing the fluorescence signals of Nub-PD (magenta), α- and γ-tubulin (Tub, green), and DNA (blue) after 0 min (control) and 60 min of cold treatment. Post-treatment recovery of 2, 5 and 10 min is indicated. (G) Schematic illustration of Nub localization after cold treatment. The panel indicates the loss or alteration of Nub enrichment around the spindle when microtubules are depolymerized, illustrating its dependency on intact spindle microtubules. Images in all panels are representative of at least three experimental repeats. Scale bars: 10 µm.

Using live imaging in cells and syncytial embryos, we show that Nub-PD localizes within the spindle envelope. This enrichment depends on intact microtubule and likely regulated by mitotic factors, including the Cdk1, Nek2, and NiKi kinases, the motor proteins Klp61F and Klp3A, and the polarity determinant Crb. Similar regulatory mechanisms have been reported for human POU/Oct proteins, including Nek6-mediated phosphorylation of POU2F1 for spindle pole localization in human cells (Kang et al., 2011) and AurB- and Cdk1-mediated phosphorylation of POU5F1 during G2/M to regulate its chromatin dissociation in embryonic stem cells (Kim et al., 2018; Shin et al., 2016). Interestingly, we found that sequence-specific DNA binding, and consequently transcriptional regulation, is not important for Nub-PD localization to the spindle or for other mitotic functions, as constructs with mutations in the NLS did not produce mitotic defects. In addition, the closely related Pdm2 paralog, which contains almost identical DNA binding POU domain and homeodomain (98% and 79% identity respectively), did not predominantly localize to mitotic spindles. Except for these domains, Nub and Pdm2 do not show regions of high amino acid identity. Taken together, these findings and the mitotic phenotypes observed in transcriptionally inactive $nub^1$, $MTD>nub^{RNAi}$ and $nub^1$;$mat>nub^{RNAi}$ embryos, strongly support the idea that Nub-PD plays a crucial non-transcriptional function in mitosis during early embryonic development in addition to its role in transcriptional regulation later during development. Similarly, our findings suggest that the human POU2F1 confers important roles in mitotic progression. Thus, Nub-PD and POU2F1 play dual roles, functioning as transcriptional regulators during interphase and in differentiated cells, and as key contributors to mitotic progression in actively dividing cells.

Understanding exactly how Nub-PD is involved in the direct regulation of mitotic progression is challenging. Our data show that both centrosome numbers and morphology remain unaffected, and instead, centrosomes frequently detached from the mitotic spindles as a consequence of spindle collapse. Despite being detached, these free centrosomes retained immunoreactivity for γ-tubulin and the capacity to duplicate, indicating that centrosome formation and integrity do not involve Nub-PD. Instead, data from syncytial embryos support a model in which Nub-PD interacts, directly or indirectly, with spindle microtubules, rather than centrosomal

microtubules or the spindle matrix, through a mutually dependent interaction. Intact microtubules and the identified mitotic proteins are required to ensure proper localization of Nub-PD within the spindles, and thereby its function in supporting proper spindle architecture and dynamics during metaphase and anaphase. Consistent with this, cold-depolymerization and recovery reveals that $nub^1$ embryos can initiate microtubule regrowth around chromosomes, but fail to convert these regrowing microtubules into a stable bipolar spindle, suggesting inefficient coupling and integration of chromosome-derived microtubules with centrosome-derived arrays leading to disorganized spindles and progressively affected midzone bundles. Together, our data suggests a model in which Nub-PD maintains spindle stability by promoting the organization and stabilization of spindle and mid zone microtubules. In a subset of embryos, the complete absence of microtubule regrowth is consistent with a role for Nub-PD in supporting efficient spindle microtubule regrowth, either by influencing early nucleation events at γ-tubulin-containing sites or by stabilizing nascent α–β-tubulin polymers during spindle reassembly. Alternatively, this phenotype might reflect impaired chromatin-mediated microtubule nucleation and/or stabilization, for instance through Ran- and/or CPC-dependent pathways, (Moutinho-Pereira et al., 2013; Petry, 2016) consistent with Nub being a chromatin-associated DNA-binding transcription factor. An important remaining question is how Nub-PD is anchored within, and persists in, the spindle apparatus. Based on our results, Nub-PD recruitment to spindle microtubules might be mediated by the motor proteins Klp61F or Klp3A and/or other microtubule-associated proteins (MAPs) not identified here. One possible model is that Nub-PD cooperates with Klp61F and Klp3A and/or specific MAPs to promote the assembly and organization of microtubule bundles, including the sliding of antiparallel microtubules required for proper spindle elongation. Recent work has shown that Klp61F physically interacts with Crb, Galla-2 and Xpd (the CGX complex) in syncytial embryos (Hwang et al., 2020). However, how the CGX complex and each component regulates Klp61F levels and activity is unclear. Loss of Nub-PD in the $nub^1$ mutant causes defects in nuclear syncytial divisions identical to those caused by Crb depletion (Hwang et al., 2020). In addition, knockdown of Crb, but not Galla-2 or Xpd, disrupts the enrichment of Nub around the spindle. These results suggest that Nub acts downstream or in parallel with Crb and the motor protein Klp61F to promote proper spindle architecture and elongation during anaphase (Fig. 8E).

It is interesting to note that many MAPs involved in spindle assembly undergo liquid–liquid phase separation and form condensates on microtubules (Sun et al., 2024). Importantly, both immunostaining and live imaging revealed that Nub-PD is present in punctate structures within the spindles (Figs 1H, 6A; Fig. S1K), raising the question of whether Nub-PD functions by acting within condensates on microtubules. Protein prediction tools for unstructured regions (IUPred2A) (Erdős and Dosztányi, 2020; Mészáros et al., 2018) and for three-dimensional structures (AlphaFold2) (Jumper et al., 2021; Varadi et al., 2024) show that outside the DNA-binding domains, Nub-PD is highly disordered, except for one coiled-coil structure (data not shown). Interestingly, POU2F1 also contains a coiled-coil structure in an otherwise primarily unstructured protein, besides the DNA-binding domains. The exact molecular mechanism of how Nub-PD promotes anaphase spindle stability and proper elongation, along with structure–function analyses, will need further investigation.

Many human POU/Oct proteins are linked to cancer development or progression, and high expression of POU proteins in malignant cells is correlated with poor prognosis (Ben-Batalla et al., 2010;

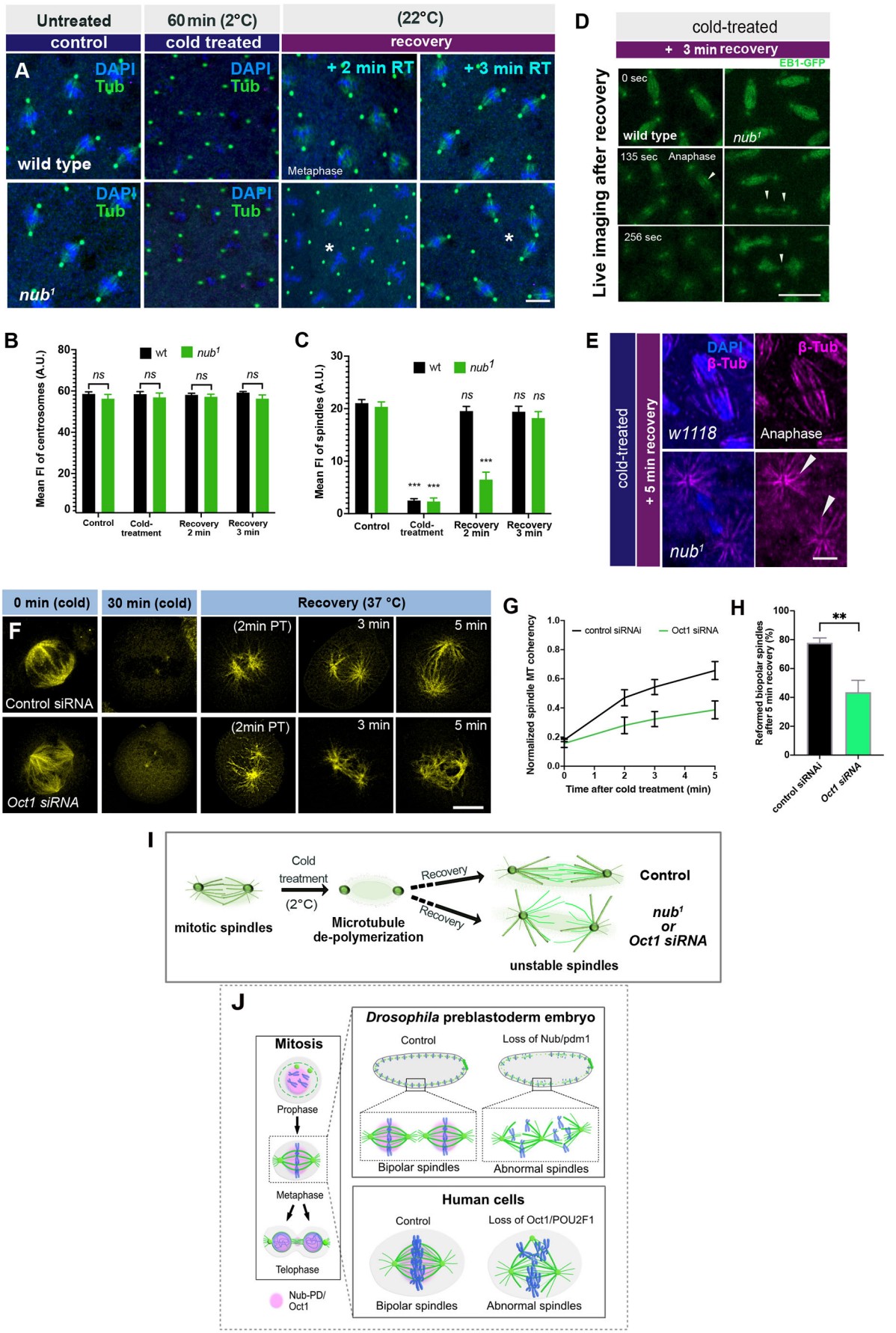

**Fig. 7.** See next page for legend.

**Fig. 7. Nub-PD and POU2F1 are required for spindle microtubule dynamics during reconstitution and regrowth.** (A) Confocal images of representative spindles in wild-type and *nub¹* embryos stained for α- and γ-tubulin (Tub, green). Embryos were cold treated for 0 min (control) and 60 min. The recovery time is indicated (min). (B,C) Quantification of centrosomal (B) and spindle (C) microtubule fluorescence intensity in wild-type and *nub¹* embryos under control conditions, cold treatment, and during MT regrowth (2–5 min recovery). Mean±s.e.m.; *n*>350 centrosomes and *N*>5 embryos per condition. ns, not significant; ***P<0.001 (unpaired two-tailed *t*-test). (D) Selected confocal frames from live imaging of dividing *wt* and *nub¹* nuclei after 3 min of recovery at room temperature following cold treatment. Arrows depict interpolar microtubules in the midzone during anaphase. (E) Confocal frames of dividing nuclei of *wt* and *nub¹* during anaphase stained for β-tubulin (magenta) and DAPI (blue). Arrowheads indicate unstable spindle bundles of *nub¹* embryos, forming aster-like structures. (F) Confocal images of representative mitotic spindles in live HeLa cells treated with control or POU2F1 (Oct1) siRNA. Cells were incubated at 0.5–2°C for 30 min (cold) and to ViaFluor® 488 to visualize spindle microtubules (yellow). Time points (in minutes) indicate post-treatment (PT) recovery duration at 37°C during mitotic progression. (G,H) Normalized spindle microtubule coherency over time after cold treatment in control versus POU2F1 siRNA-treated cells, showing impaired MT reassembly upon POU2F1 knockdown (G). Spindle MT organization was quantified from α-tubulin images using the OrientationJ coherency index in Fiji/ImageJ, which reports local filament alignment (0=random, 1=perfectly ordered) (see Materials and Methods for details). Coherency over time in control versus POU2F1 siRNA was scientifically different (*P<0.01*), as obtained by a logistic regression model using the Chi-squared test. Percentage of cells that reformed bipolar spindles 5 min after recovery, demonstrating a significant reduction in POU2F1 -depleted cells (H). Mean±s.e.m., (values are a sum of three independent experiments, ≥60 cells per condition); **P<0.002 (unpaired two tailed *t*-test). (I) A schematic representation showing the microtubule regrowth in wild-type (control) and *nub¹* or POU2F1/Oct1-siRNA after cold treatment. Microtubule regrowth was aberrant after cold treatment in both *nub¹* mutants and POU2F1 siRNA knockdown, leading to unstable spindle formation. (J) Schematic summary of the observed phenotypes showing microtubule regrowth defects in syncytial *Drosophila* embryos and human cells. Loss of Nub (Pdm1) leads to unstable microtubule bundles within the spindle, resulting in misorientation of the division axis and aberrant interactions with neighboring spindles. In human cells, downregulation of POU2F1 disrupts spindle organization and bipolarity, leading to the formation of extra spindle poles. Scale bars: 10 μm.

Castrillo et al., 1991; Jullien et al., 2015; Rudin et al., 2019; Vázquez-Arreguín et al., 2019; Vázquez-Arreguín and Tantin, 2016). In fact, POU2F1 is expressed in all human tumor cell lines and it has been found to be proto-oncogenic in epithelial malignancies (Jafek et al., 2019; Stepchenko et al., 2022; Vázquez-Arreguín et al., 2019; Vázquez-Arreguín and Tantin, 2016). However, the molecular role of POU2F1 in tumor proliferation remains unclear. We show in live HeLa and HEK293 cells that POU2F1 is crucial for the normal progression of mitosis and that its downregulation results in spindle collapse leading to prolonged mitotic progression, and the formation of multinucleated cells. These results correlate with earlier findings in fixed HeLa cells, showing that both up- and down-regulation of POU2F1 causes abnormal mitosis (Kang et al., 2011). The phenotypes we observe upon POU2F1 depletion might be explained by distinct mechanisms in which centriole amplification or fragmentation could drive spindle multipolarity, while cytokinesis failure could give rise to multinucleation. Both scenarios remain possible, and additional experiments will be required to distinguish between centrosome-related defects and cytokinesis failure in this context. Our live imaging in a cold-induced microtubule regrowth assay demonstrates that POU2F1 is required for maintaining spindle architecture and promoting rapid spindle microtubule regrowth following microtubule de-polymerization. This phenotype is remarkably similar to that observed upon loss of the *Drosophila* Nub-PD isoform, supporting a

similar and potentially conserved function of POU/Oct factors in stabilizing spindle microtubule organization. This points at new roles for POU/Oct proteins in cellular proliferation and tumorigenesis, however, the molecular mechanism underlying this mitotic function and its broader relevance for tumor progression remain to be elucidated. Whether this mitotic role of POU2F1 is mechanistically distinct from its transcriptional activity also remains to be determined.

Our results suggest an important contribution of POU2F1 and Nub-PD in efficient spindle organization and chromosome segregation in systems with fast divisions, such as tumor cells and syncytial embryos. These so far overlooked additional roles of POU/Oct proteins in mitosis might explain some of the unanswered questions regarding their oncogenic properties, and open new possibilities in drug targeting and pharmaceutical interventions to block cell proliferation and malignant growth.

## MATERIALS AND METHODS
### Experimental model details – *Drosophila melanogaster*
Flies were grown on instant potato and yeast food (Dantoft et al., 2016) at 25°C, and experiments were performed at 25°C unless otherwise specified. *Drosophila melanogaster* stocks used in this study and associated references are listed in the key resources table (Table S2). Some stocks were treated with tetracycline for two to three generations to be cleaned from *Wolbachia* infection, which then was verified by PCR.

### *Drosophila* genetics
Details for the source of the genotypes used are in the key resources Table S2. CyO and TM3 balancer strains carrying GFP transgenes were used to identify embryos with the desired genotypes for fly genetics.

### Germline-specific RNAi knockdown
Transgenic flies carrying RNAi vectors (Valium20 or Valium22), specifically designed for optimal expression of RNAi hairpins in the germline (Blake et al., 2017) were used. Homozygous males from the RNAi lines were mated with virgin females of the Maternal Triple Driver (MTD)-Gal4, which drives expression throughout oogenesis. *MTD>shRNA* females from this cross were backcrossed to the homozygous *UAS-shRNA* or to F1 *MTD>shRNA* males. Their F2 embryo progenies were analyzed. Flies were kept at 25°C continuously. We aimed for partial downregulation conditions in this experiment because strong depletion of important mitotic regulators can cause global mitotic collapse, which would prevent meaningful interpretation of Nub-PD localization.

### Nub-PD degradation in the early embryo using deGradFP system
To degrade Nub-PD protein in early embryos, *nos>deGradFP* (Gaskill et al., 2021) and *hb>deGradFP* (Vazquez-Pianzola et al., 2022) transgenic lines were used. Briefly, *nos>deGradFP* flies were crossed with *sfGFP-nubPD* and offsprings were selected for nos>*deGradFP/sfGFP-nub-PD* and intercrossed. Similarly, *hb>deGradFP* flies crossed with *sfGFP-nubPD* and offsprings were selected for *sfGFP-nubPD*; *hb>deGradFP,* and intercrossed. From these crosses, embryos (0–2 h) were collected on apple juice-agar plates supplemented with yeast at 25°C. Embryos from *nos>deGradFP* and *hb>deGradFP* females were used as a control.

### *D. melanogaster* cell culture and RNA interference
*D. melanogaster* Schneider's (S2) cells, and S2 cells that stably express Histone 2B–GFP (H2B–GFP) and α-tubulin–mCherry were maintained in Schneider's insect medium (Gibco; cat. no. 21720024), supplemented with 10% heat-inactivated fetal bovine serum (FBS, Gibco, cat. no. A5670801) and 1% penicillin-streptomycin antibiotics cocktail (Gibco, cat. no. 15070063) at 25°C without $CO_2$.

Double-stranded RNA (dsRNA) targeting firefly luciferase (*Luc*; control) and *nub* transcripts (*nub-RB* and *nub-RD*) were synthesized using a T7 RiboMAX™ Large Scale RNA Production Systems kit (Promega, #P1300) according to the manufacturer's instructions. S2 cells were incubated with respective dsRNA for 48–72 h at 25°C.

Journal of Cell Science

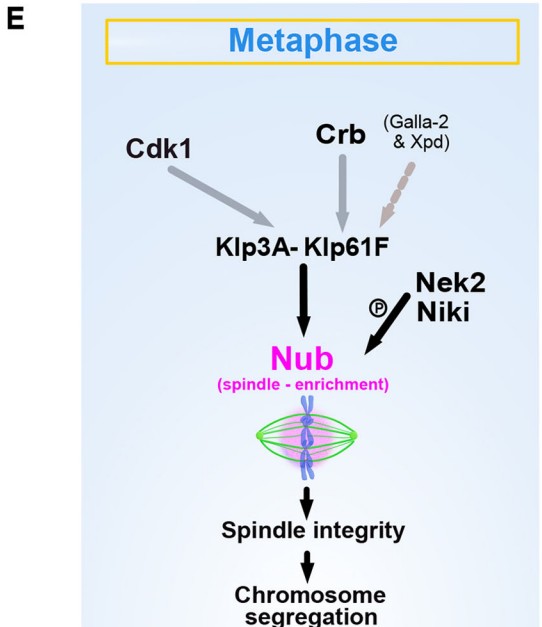

**Fig. 8. Factors affecting Nub-PD recruitment to mitotic spindles.** (A–D) Airy-scan confocal images displaying the cortex of *MTD>RNAi^GFP* (control) (A), *MTD>RNAi^Deterin* (B), *MTD>RNAi^Incenp* (C) and *MTD>RNAi^Klf61F* (D) syncytial embryos stained with antibodies for Nub (magenta), β-+γ-tubulin (green) and DAPI (blue). Mitotic phases of nuclei are indicated (interphase, metaphase). Images are representative of at least three experimental repeats. Scale bar: 10 μm. (E) Proposed model of Nub-PD recruitment to the mitotic spindle, incorporating findings from both current (black) and previously published work (gray). The recruitment of Nub-PD to the mitotic spindle involves the mitosis-related motor proteins Klp61F and Klp3A, the polarity determinant Crb and the kinases Cdk1, Nek2, and NiKi/Nek9. The enrichment of Nub around the spindle supports spindle integrity and accurate chromosome segregation.

### Human cell culture and siRNAs

HeLa cells were maintained in DMEM F12 or DMEM GlutaMax (Gibco) media supplemented with 10% heat-inactivated FBS (Gibco) at 37°C with 5% $CO_2$. Cells were transfected with POU2F1 siRNAs, targeting three independent regions of human POU2F1 (sc-36119) and control siRNA (sc-37007) (Santa Cruz Biotechnology), at 100 nM siRNA concentrations. Transfections were carried out with Lipofectamine-RNAiMAX reagent (Life Technologies) following the manufacturer's instructions.

### Molecular biology and transgenics

#### Generation of *sfGFP-nubPB* and *sfGFP-nubPD* flies by CRISPR/Cas9-mediated homology-directed repair

##### Design and construction of gRNA plasmids

We used the CRISPR optimal target finder algorithm (https://flycrispr.org/target-finder/) to identify suitable genomic target sequences in the first protein-coding exons of *nub-RB* (*nub* exon 2) and *nub-RD* (*nub* exon 4). The following CRISPR genomic target sites were selected for targeting Nubbin isoforms with GFP: Nub-RB.Trg1, 3′-GGATGCTGAAACGCCTTCGG-**TGG**-5′; Nub-RB.Trg2, 3′-CGGATGATGATACAGCAGTG**TGG**-5′; Nub-RD.Trg1, 3′-TGGTTATGTCGGAGCTACGT**TGG**-5′; and Nub-RD.Trg2, 3′-TTGGCACACCGCTAGTCCCG**AGG**-5′ (PAM sequences are highlighted in bold). Target sequence-specific sense and antisense oligonucleotides (Table S2) with 5′ complementary overhanging sequences to the BbsI-cut *pCFD3-dU6:3gRNA* vector were PNK-phosphorylated and annealed prior to ligation into BbsI-cut *pCFD3-dU6:3gRNA* (http://crisprflydesign.org/protocols/). The resulting guide plasmids were confirmed by Sanger sequencing (Eurofins Genomics).

##### Construction of pBS II SK(+)-Nub-RD.sfGFP and pBS II SK(+)-Nub-RB.sfGFP donor vectors

Approximately 1-kb long DNA sequences upstream and downstream of the genomic target sites were PCR-amplified from fly genomic DNA (BL#58492) with the Q5 DNA polymerase kit (NEB, #M0491S) using Nub-PD.HL.Fw and Nub-PD.HL.Rw, or Nub-PD.HR.Fw and Nub-PD.HR.Rw primers that added flanking restriction sites for *XhoI* and *HindIII* (Nub-PD.HL), or *HindIII* and *BamHI* (Nub-PD.HR) to the PCR products. XhoI/HindIII-digested Nub-PD.HL, and HindIII/BamHI-digested Nub-PD.HR were ligated in the XhoI/BamHI-cut *pBluescript* (referred to as *pBS*) II *SK(+)* vector. Nub-PDsfGFP.Fw and Nub-PDsfGFP.Rw primers were used to PCR-amplify the sfGFP-coding sequence from the *pScarlessHD-sfGFP-DsRed* vector (Addgene #80811), simultaneously adding 25 bp flanking sequences that overlapped with the Nub-PD.HL/Nub-PD.HR-hinge region. The resulting PCR product was inserted into HindIII-linearized *pBSII SK(+)-Nub-PD.HL/Nub-PD.HR* with the HiFi DNA assembly kit (NEB, #E2621S). Similarly, we constructed *pBS II SK(+)-Nub-RB*.sfGFP using ~1 kb homology-providing sequences flanking the genomic target sites that were PCR-amplified from fly genomic DNA (BL#58492) with the primers Nub-PB.HL.Fw and Nub-PB.HL.Rw, or Nub-PB.HR.Fw and Nub-PB.HR.Rw adding flanking *XhoI/HindIII* or *BamHI/NotI* restriction sites. XhoI/HindIII-, respectively BamHI/NotI-cut, PCR-products were ligated into the respectively digested *pBS II SK(+)* vector. Nub-RB.sfGFPFw and Nub-RB.sfGFPRw primers were used to amplify the *sfGFP* CDS from *pScarlessHD-sfGFP-DsRed* simultaneously adding 25 bp flanking sequences that overlapped with the Nub-PB.HL/Nub-PB.HR-hinge region. The *sfGFP*-coding fragment was inserted into EcoRV/BamHI-opened *pBS II SK(+)-Nub-PB.HL/Nub-PB.HR* via HiFi DNA assembly (NEB). The final *pBS II SK(+)-Nub-RD.sfGFP* and *pBS II SK(+)-Nub-RB.sfGFP* donor constructs were verified by Sanger sequencing (Eurofins Genomics).

##### Fly injections and candidate screening

The Cas9-expressing fly line *y1,M[Act5C-Cas9]ZH-2A, w1118, DNAlig4169* (BL#58492) was used for embryo injections. A mix of two *pCFD3-dU6:3gRNA* guide vectors and the respective donor construct was co-injected into fly embryos. Each resulting $G_0$ individual was crossed to *w1118* flies, and subsequent screening for GFP expression was done in the adult $G_1$ population. GFP-positive $G_1$ individuals were crossed to a *CyO* balancer stock, and the presence and correct integration of sfGFP were further confirmed by PCR on isolated genomic DNA. The Cas9 carrying allele *y1, M[Act5C-Cas]}ZH-2A, w1118, DNAlig4169* located on the first chromosome was segregated away. Homozygous *nub^{sfGFP.Nub-PB}* and *nub^{sfGFP.Nub-PD}* fly stocks could be established and were further verified by genomic DNA sequencing (Eurofins Genomics), and analyzed for viability, life span, and fecundity. In *nub^{sfGFP.Nub-PB}* flies, the DNA sequence coding for Val2-Lys239 of sfGFP is inserted between 2L:12,594,048 and 2L:12,594,049 [+], resulting in an N-terminal insertion of the sfGFP tag following Thr17 of the predicted Nub-PB protein. In the case of *nub^{sfGFP.Nub-PD}* flies, sfGFP (Val2-Lys239) is inserted n-terminally after Leu6 of the predicted Nub-PD isoform (between 2L:12,619,013 and 2L:12,619,014 [+]). In both cases, the sfGFP coding sequence contains a short C-terminal extension (coding for Ile-Gln-Pro-Arg-Lys-Ile-Ile) originally present in the *pScarlessHD-sfGFP-DsRed* plasmid.

### Antibody production and immunostaining

Antibodies against Pdm2-PA/PB isoforms were raised in rabbits (Agrisera AB, Vännäs, Sweden) against a synthesized peptide (PPKRLAEEQEEEK) conjugated to keyhole limpet hemocyanine carrier protein (Thermo Fisher Scientific, Waltham, MA, USA). The Pdm2 peptide without the carrier protein was coupled to cyanogen bromide-activated Sepharose 4B according the manufacturer's protocol (Sigma-Aldrich, St Louis, MI, USA), and used for affinity purification of the antisera.

#### Embryo

Embryos were bleached, dechorionated, and fixed for 20 min in 4% formaldehyde (or for 15 min in 0.5% formaldehyde) and incubated overnight at 4°C with primary antibodies. Secondary antibodies conjugated to Cy3, Cy5, Alexa Fluor 488 or Alexa Fluor 568 were diluted as recommended by the manufacturers and embryos were incubated for 2 h at room temperature. Stained embryos were mounted in a Vectashield Plus antifade mounting medium.

#### S2 and HeLa cells

Cells were seeded and fixed in 4% fresh paraformaldehyde for 30 min. Fixed cells were blocked in PBS with 0.1% Triton X-100 (PBST) supplemented with 0.5% normal goat serum (NGS) for 60 min at room temperature and incubated with primary antibody overnight at 4°C. The next day, cells were washed and incubated with a secondary antibody for 2 h at room temperature. DNA was counterstained with DAPI. Cells were mounted in 90% glycerol or in Vectashield Plus antifade mounting medium for imaging.

### Injections of anti-Nub and imaging

Manually dechorionated embryos were placed on coverslips coated with a heptane–glue mix and then covered with halocarbon oil 700. Monoclonal mouse anti-Nub (Nub 2D4) was diluted in injection buffer (50 mM HEPES pH 7.4 and 50 mM KCl). Nub antibody (12 µM) was spun at 13,000 *g* for 15 min (4°C) and the supernatant was co-injected with 3 kDa Alexa Fluor™ 488 dextran (10 mg/ml) into syncytial embryos (NC8) using a microinjection system (FemtoJet, Eppendorf) coupled to an inverted fluorescence microscope (Cell Observer, Zeiss). Control embryos were injected with injection buffer only, containing the same concentration of dextran. Injections were performed at the anterior part of the embryo. The injected embryos were immediately placed to an AxioImagerZ1 (Zeiss) microscope for live imaging by using a 20×/0.75 NA Plan-APOCHROMAT objective (Carl Zeiss).

### Live imaging

#### Embryo

Embryos (0–1 h old) were dechorionated for 3 min in 7% sodium hypochlorite solution and mounted onto a pre-prepared slide (Tsarouhas et al., 2019). For wide-field live imaging, embryos were imaged with a CCD camera (AxioCam 702, Carl Zeiss) attached to an AxioImagerZ1 (Zeiss) microscope by using a 20×/0.75 NA Plan-APOCHROMAT objective (Carl Zeiss). Individual Z-stacks with a step size of 1.2–1.9 µm were taken every 1 or 2 min over a 2–3.5 h period. Embryos were collected from yeasted grape juice agar plates aged at 25°C. For confocal imaging, embryos were imaged with a laser-scanning confocal microscope (LSM 780, Carl Zeiss) using a 63× Plan-APOCHROMAT/1.4 M27 oil-immersion objective. For high-resolution confocal imaging, an airy-scan-equipped confocal microscope

system (Zeiss LSM 800, Carl Zeiss) with a 63× Plan-APOCHROMAT/1.4 M27 oil immersion objective was used. Raw data were processed with the airy-scan processing tool of the Zen Black software version 2.3 (Carl Zeiss). Images were converted into tiff format using ImageJ/Fiji software (http://rsb.info.nih.gov/ij/).

### S2 and HeLa cells
For analyzing the mKO2- and RFP-tagged Nub-PD protein localization, S2 cells were seeded into 35 mm glass-bottom cultured plates (MatTek) overnight. Cells were transfected with 100 μg of pW8 (carrier plasmid) using Effectene transfection kit (Qiagen). After 48 h of transfection, cells were treated with ViaFluor 488 for 20 min to visualize the microtubules. Images were taken every 1 min for a period of 2-4 h. For mitotic phenotypes, S2 cells that stably expressed Histone 2B–GFP (H2B–GFP) and α-tubulin–mCherry treated with control or *nub-RD* dsRNA for 72 h. For cell imaging, Zeiss LSM 800 Airy-scan confocal microscope with a Plan/Apo 63×1.4 NA oil immersion objective was used. Images were analyzed using Zeiss 2011 (Blue) software and were exported in tiff format. The tiff files were imported into ImageJ/Fiji software to generate the videos.

### RNA extraction and RT-qPCR
Embryos were collected for 2 h, and subsequently washed and dechorionated. Syncytial blastoderm-staged embryos were collected in Trizol. S2 cells were harvested from six-well plates, washed with PBS, and resuspended in TRizol. RNA extraction and RT-qPCR were performed as previously described (Lindberg et al., 2018).

### Cold-induced microtubule de-polymerization
For cold-treatment assays, embryos (0–2 h) were collected, dechorionated, placed on ice-cold Petri dishes, and incubated on ice inside polystyrene boxes in a 4°C incubator room. Following 60 min treatment on ice, embryos were fixed at 0 s (immediately after cold treatment and inside the cold room), 60 s, 120 s, 180 s, 5 min or 10 min after shifting them to 25°C. For live imaging, dechorionated embryos were mounted onto a slide with a gas-permeable membrane and covered with halocarbon oil (#700, Sigma) (Tsarouhas et al., 2019). Slides were placed on ice-cold Petri dishes and incubated on ice for 60 min, before quick transfer to the microscope (within 30–60 s) and were imaged immediately, focusing on the centrosomes (Conduit et al., 2015; Hayward et al., 2014). Similarly, human cells growing in culture in glass bottom dishes were incubated on ice for 30 min inside polystyrene boxes at in a 4°C incubator room. The treatment duration was optimized in pilot experiments to induce spindle microtubule de-polymerization in over 85% of cases without impairing the capacity for regrowth. A digital thermometer, equipped with probes attached inside the dishes, continuously recorded the temperature, which was constantly kept at 2.0±0.5°C.

### Quantifications and statistical analysis
#### Quantifications of nuclei dynamics
Confocal micrographs were transformed into the 'binary' images followed by the Fiji/ImageJ plugin 'Segmentation Editor' image for threshold setting and automatic calculation. The relative nuclear to cytoplasmic ratio (N/C ratio) was calculated on binary images as: $N/C=E_{nu}/(E_{cyto}=E_{total}-E_{nu})$, where $E_{nu}$=area of nuclei, $E_{cyto}$=area of the cytoplasm and $E_{total}$=area of the whole embryo (the visible part). $E_{nu}$ was defined by the His2A–RFP or DAPI signals after manually tracing the borders of nuclei. $E_{total}$ was estimated by the fluorescence signal of the whole embryo. Cortex area with nuclei gaps was calculated after tracking the cortical area without nuclei (gaps) and dividing this area by the area of the whole embryo.

For the time profile of mitotic phase in nuclei of syncytial embryos, S phase was defined as the period from early interphase to the late prophase, when chromosomes just begin to condense and become visible (Yu et al., 2000). M phase includes the time of metaphase (chromosomes in the equator) and anaphase until early telophase (Fasulo et al., 2012).

#### Quantifications of RFI and spindle dynamics
Image processing and quantification were performed using Fiji software. Confocal images were exported in *tiff* format (uncompressed) by Zeiss 2011 (Blue) software. The relative fluorescence intensity (RFI) of Nub was calculated by dividing the average fluorescence intensity of Nub signals in the nucleus (Fln, in interphase) or spindles (Fls, in metaphase) by the fluorescence intensity of the cytoplasm (FIc) outside the nucleus or the spindle area. $RFI=FI_n/FI_c \ or \ FI_s/FI_c$. The fluorescence intensity was measured using a region of interest (ROI) defined by the limits of the nuclei in interphase (DAPI signal) or by the spindles in metaphase (tubulin signal). The background was measured with the equivalent ROI outside the area of the nuclei or the spindle. Images were sum-intensity Z-projections.

The pole-to-pole distance is the length of a line (μm) drawn between the two spindle poles, defined by the centrosomes. For the pole-to-pole distance as a fraction of time (μm/s), the length of the line between the spindle poles was calculated at each time point based on EB1–GFP real-time imaging data. The quantification was performed using Fiji software and the 'Profile module' of the Zen2011 program.

#### Quantifications of centrosome fluorescence intensity and density
Centrosome density was determined by counting centrosomes within 50×50 μm regions of interest and normalizing to the ROI area (centrosomes/μm²). At least three cortical regions were selected for each embryo. For quantification of centrosome intensity, maximum-intensity projections were generated and individual centrosomes were segmented as diffraction-limited spots using Fiji. For each centrosome, a fixed-size circular region of interest (ROI; corresponding to 0.5–0.7 μm diameter) was placed over the centrosome signal, and the mean fluorescence intensity was measured after subtraction of the local cytoplasmic background (measured in a nearby ROI lacking centrosome signal).

#### Quantifications of microtubule organization
To quantify spindle microtubule organization (related to Fig. 7G), we analyzed α-tubulin fluorescence using the OrientationJ plugin (v3.0, BIG-EPFL, Fiji). Maximum-intensity projections from images were background-subtracted (rolling-ball radius 30–50 px) using identical settings for all samples. A fixed spindle ROI was defined manually for each cell and used for all corresponding timepoints. OrientationJ was run in 'Structure Tensor' mode. The plugin produced a coherency image in which pixel intensities (0–1) represent the local degree of filament alignment. For each frame, the mean coherency value within the spindle ROI was measured using Analyze and Measure tool. Result tables were exported as CSV files for subsequent plotting in GraphPad Prism.

#### Statistics
Error bars show means±s.e.m. except for in Fig. 4G. Graphs with box plots show the median (horizontal line) and the range from 25th to 75th percentile. The whiskers depict maximum and minimum values. *P*-values were calculated through the *t*-test for unpaired variables if the data passed normality test (Shapiro-Wilk test). For grouped data, one-way or two-way ANOVA followed by a Tukey's post hoc test for multiple comparisons was applied. For spindles dynamics (Fig. 4I,J), data analysis was performed using JMP Pro 17.0 statistical program (SAS Institute). Normality was determined by the D'Agostino–Pearson normality test, using a significance cut-off value of *P*=0.05.

The *n* values of the quantifications are provided in the figure legends. All plots were made with and statistical tests values calculated used GraphPad Prism 10.1 except the analysis in Fig. 4I,J (see above).

### Acknowledgements
Stocks obtained from the Bloomington Drosophila Stock Center (NIH P40OD018537) were used in this study. A monoclonal Nub antibody used in this study was developed by Michalis Averof and Steve Cohen, and was obtained from the Developmental Studies Hybridoma Bank, created by the NICHD of the NIH and maintained at The University of Iowa, Department of Biology, Iowa City, USA. We thank Cai Yu, Xiaohang Yang, and William Chia for the Pdm1/Nub antibody, and Georg Wolfstetter for critically reading and commenting on the material and methods section. The authors also acknowledge the technical support from the IFSU (Imaging Facility at Stockholm University).

### Competing interests
The authors declare no competing or financial interests.

**Author contributions**
Conceptualization: P.G., V.T., Y.E.; Formal analysis: P.G., V.T., L.K., S.S.; Funding acquisition: V.T., Y.E.; Investigation: P.G., V.T., L.K., S.S.; Methodology: P.G., V.T., L.K., S.S.; Project administration: Y.E.; Supervision: Y.E.; Visualization: P.G., V.T., L.K.; Writing – original draft: P.G., V.T., Y.E.; Writing – review & editing: P.G., V.T., L.K., Y.E.

**Funding**
This work was financially supported by The Swedish Cancer Society (Cancerfonden; 20 1044 PjF and 23 2963 Pj) to Y.E., The Swedish Research Council (Vetenskapsrådet; 2018-04401) to Y.E. and by O. E. och Edla Johanssons Vetenskapliga Stiftelse and Magnus Bergvalls Stiftelse to V.T. Open Access funding provided by Stockholm University. Deposited in PMC for immediate release.

**Data and resource availability**
Fly-lines and plasmids generated in this study are available upon request from the lead contact. All relevant data and details of resources can be found within the article and its supplementary information.

**Peer review history**
The peer review history is available online at https://journals.biologists.com/jcs/lookup/doi/10.1242/jcs.264165.reviewer-comments.pdf

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
