## [Peer Review File · Journal of Cell Science]

A non-transcriptional mitotic function of POU/Oct factors ensures spindle assembly and chromosome segregation

Priya Gohel, Vasilios Tsarouhas, Laveena Kansara, Suresh Sajwan and Ylva Engström
DOI: 10.1242/jcs.264165

Editor: Renata Basto

Review timeline

Original submission:	25 May 2025
Editorial decision:	16 July 2025
First revision received:	16 December 2025
Editorial decision:	6 January 2026
Second revision received:	5 February 2026
Accepted:	9 February 2026

Original submission

First decision letter

MS ID#: jcs.264165

MS TITLE: A non-transcriptional role of POU/Oct factors in mitotic progression

AUTHORS: Ylva Engström; Priya Gohel; Vasilios Tsarouhas; Laveena Kansara; Suresh Sajwan

ARTICLE TYPE: Research Article

Dear Dr Engström,

We have now reached a decision on the above manuscript.

As you will see, the reviewers raise a number of substantial criticisms that prevent me from accepting the paper at this stage. They suggest, however, that a revised version might prove acceptable, if you can address their concerns. If you think that you can deal satisfactorily with the criticisms on revision, I would be pleased to see a revised manuscript. We would then return it to the reviewers.

Reviewer 1

Advance summary and potential significance to field

This study by Sajwan and colleagues investigates the non-transcriptional roles of Nub/Pdm1 in mitotic spindle assembly and dynamics. The topic is both timely and challenging, as it requires disentangling potential indirect transcriptional effects from direct, transcription-independent functions.

The manuscript presents a large body of data and employs several innovative tools. However, some of the findings are correlative and do not directly support the central claims of the paper. These issues could be addressed either by additional experiments or by significantly tempering the conclusions, or by limiting the number of experiments shown. I think the authors should present fewer things but present them better with the right controls. The multiplication of experiments in

different systems (particularly in cultured cells) where the effect on transcription cannot be excluded does not necessarily help the message they want to get across. Despite these concerns, I believe that the work has potential and could be suitable for publication in Journal of Cell Science after major revision and rewriting.

Major points:

* The use of deGrad-GFP or RNAi in embryos relies on maternal drivers. The authors argue that transcripts are only translated after egg activation, but this is an oversimplification. Disrupting mitosis during embryogenesis does not preclude prior effects during oogenesis, especially on the production of mitotic components by a known transcription factor. Therefore, many of the experimental strategies used may not be fully sufficient to draw conclusions about direct mitotic roles. The authors should adopt a more cautious interpretation to avoid misleading statements. In my view, the most convincing strategy is the antibody injection during syncytial divisions, which allows a rapid and stage-specific depletion. These experiments support the hypothesis of a direct role for Nub in spindle assembly and are a strength of the study.

* Figure 1 and line 140: How can the mitotic index be decreased while metaphase/mitosis is delayed? These findings are contradictory and should be clarified.

* The efficiency of RNAi knockdown is not demonstrated. No Western blots are shown to validate depletion.

o Line 224: The absence of staining does not necessarily indicate the absence of protein. Have the authors performed Western blots to support this?

* Line 252: The claim of defective chromosome segregation should be supported by appropriate live imaging, which is not convincing at all (e.g., Figure 2D). This is strange the authors decided to show these panels as nice movies are provided.

* Line 352: What is the evidence for spindle "degradation"? Is this term appropriate?

* Line 308: Why would spindle dynamics be abnormal in controls?

* Line 314 and 548: Centrosome detachment from the spindle is a known defect, but is it the spindle that detaches from centrosomes or vice versa? Please clarify.

* As mentioned above, conclusions from the deGrad-GFP phenotype are only valid if Nub-sfGFP is not degraded during oogenesis (It should control transcription of essential genes). A Western blot of Nub-sfGFP-D with and without the degradation driver during oogenesis would help determine whether the observed effects are truly embryo-specific. This control would also support better the antibody injection experiments and the "transcription independent effect" on mitosis.

* Is the Nub-GFP-PD construct functional? Does it rescue the nub1 phenotype? It remains unclear whether the nub1 phenotype is due to loss of PB or PD isoforms. This distinction should be clarified, particularly if the authors aim to assign a specific role to Nub-PD in the MT regrowth assay.

* Figure 8, line 522: It is well known that altering spindle morphology or MT density (e.g., via nocodazole or colchicine) can affect the localization of MT-associated proteins. For instance, Incenp RNAi disrupts the spindle, so reduced Nub localization is expected. These measured effects may be indirect and do not demonstrate direct regulation of Nub by CPC. The authors could assess the Nub:MT signal ratio to strengthen this argument, but given the weak Nub spindle association and poor tubulin staining in some images, this might be technically challenging. I suggest removing the speculative recruitment model in Figure 8E and significantly softening the conclusions drawn from these experiments (see also discussion, lines 568-572).

* There is confusion regarding Nub localization "around the spindle." Given that the spindle is compartmentalized by membranes (as shown in works by Maiato, Baum, and others), the current terminology is misleading. In embryos, Nub1 appears around chromosomes, whereas in S2 cells, it co-localizes with the spindle.

* The interpretation of the MT regrowth assay is unclear and should be more discussed.

* All figures and movies must include time stamps and scale bars. This is a minimal requirement for publication in a cell biology journal, yet many figures lack this.

* Fixation may mask epitopes. Could the authors provide live imaging movies of sfGFP-Nub isoforms with RFP- or mCherry-tubulin?

* S2 cells (Figure S1 C-H): S2 cells express both isoforms, but RNAi only achieves ~50% knockdown of the 2 (B and D) RNAs. No Western blots are provided, which raises concerns about the reliability of the reported experiments. Additionally, I am surprised that the authors use midbody staining in cells at various stages of division (from telophase to abscission) to assess protein depletion. This approach is unconventional and not standard practice in the field.

- * Figure S1 I-L: The authors suggest Nub function is conserved in human cells, but the phenotypes are poorly characterized. Multipolar spindles may result from centriole amplification or fragmentation, whereas multinucleation (as indicated in the graphs) implies cytokinesis failure. No images of interphase polyploid cells are shown to sustain that hypothesis.
- * The MT regrowth assay suggests that chromosome-driven MT nucleation (regulated by Ran/CPC) is affected by Nub. As Nub contains an NLS, it may be relevant to discuss possible connections with Ran-dependent pathways. Interestingly, centrosome-based MT nucleation increases in Nub mutants during anaphase, while central spindle MTs decrease. Can the authors explain this phenotype?
- * Figures 6 and 7 appear somewhat redundant. Is the function of human Oct1 different from that of Nub? The phenotypes differ. This should be addressed.
- * Figure 5C-D: It is unclear whether these panels represent different treatments. Please clarify in the legend.
- * Figure 3: Pole-to-pole distance decreases in Nub1 mutants at metaphase in cycle 9 but remains unchanged during pre-anaphase B in cycle 10. Is the phenotype cycle-specific?
- * Figure 2D: On what basis do the authors claim these are anaphase figures in Nub1 mutants?
- * Figure 2E-G: The measurement of cell cycle phases is surprising given the poor image quality. I would therefore suggest to show better images with a higher magnification. NEBD cannot be assessed without nuclear envelope markers. Alternatively, nuclear entry of cytoplasmic proteins in the nuclear space could be used. Without this, the measurements are unreliable. I strongly recommend showing only data supported by the H2B nuclear marker (if that is what was used). Otherwise, these data should be removed. At least the authors could explain how the data were obtained in the material and Method section.
- * Figure 1I: What exactly is plotted? How are values normalized?
- * No Western blots are shown to validate the new anti-Pdm2 antibodies.

Unsupported claims:

- * Line 523: "This spindle enrichment is microtubule-dependent." No data are shown to support this.
- * Line 532: "...and thereby for its mitotic functions." This is not demonstrated.
- * Line 543: "Thus, Nub-PD and POU2F1/Oct1 play dual roles..." This is speculative for the human ortholog. As the authors themselves state in line 609-610, "Whether this mitotic role of POU2F1/Oct1 is mechanistically distinct from its transcriptional activity remains to be determined."
- * Line 547: "Our data show that both centrosome numbers and morphology remain unaffected." Yet, centrosome numbers are not quantified. Multipolarity could result from cytokinesis defects or centriole amplification. Please amend this sentence or provide data.
- * Line 558: "...supporting proper spindle architecture and dynamics during metaphase and anaphase." Why not earlier mitotic stages?
- * Line 600: "...and the formation of multinucleated cells." Please provide images or FACS analysis to support this, or remove the claim.
- * Line 604: "...promoting microtubule regrowth in human cells." This is not evident from Figure 7E, which is also not quantified.

The data does not reflect a role on mitotic progression (as mentioned in the title) but on spindle assembly and chromosome segregation.

Reviewer 2

Advance summary and potential significance to field

This study revealed that *Drosophila* Nub1/Pdm1 and its human homolog POU2F1/OCT1 play a role in ensuring accurate mitosis, which is distinct from their well-known function in transcription.

Nub1 has two isoforms, Nub1-PB and Nub1-PD in *Drosophila*. siRNA-mediated knockdown of Nub1-PD caused deformation of the mitotic spindle in *Drosophila* S2 cells (Fig. 1A-C). Similarly, siRNA against OCT1 (siOCT1) induced mitotic defects in human HeLa cells (Fig. 1D-E). In *Drosophila* embryos, only Nub1-PD is expressed, and zygotic transcription is not required for the syncytial division cycles. Under these conditions, nub1 mutants exhibited mitotic defects, and the embryo cortex was not fully covered by nuclei (NUF phenotype; Fig. 2). As observed in S2 cells, nub1 mutant embryos

showed disorganized mitotic spindles, prolonged mitosis, and shortened spindle length (Fig. 3). The NUF phenotype and spindle defects were rescued by introducing an additional copy of the nub1 gene (Fig. 4). Protein knockdown of GFP-fused Nub1-PD using deGradFP caused even more severe mitotic defects in embryos. Nub1-PD localized around chromosomes and spindle bundles during metaphase in a tubulin-dependent manner (Fig. 6). The spindle localization of Nub1-PD depends on Nek2, Niki, the CPC complex, Klp61F, Klp3A, and Crb (Fig. 7).

Overall, the study provides convincing evidence that Nub1-PD plays a role in proper mitotic spindle formation, which is distinct from its function in transcription. I did not find a major problem for publication. However, the authors should consider the following minor points in a revised version.

Line 156: Fig. SG-H should be corrected to Figure S1G-H.

Fig. 5E-G: the presented images show clear defect in mitosis at NC9. However, it is more convincing to include a quantified data as well. Additionally, is it possible to observe the defect at earlier NC stages?

Lines 424-427: Megator showed a similar localization pattern as Nub-PD in wild type embryos, and this did not change in nub1 mutant embryos. In contrast to Nub-PD, Megator remained intact in a mesh-like structure around the spindles after cold treatment (Fig. S5D). These sentences should be as follows.

Megator showed a similar localization pattern as Nub-PD in wild type embryos, and this did not change in nub1 mutant embryos (Fig. S5D). In contrast to Nub-PD, Megator remained intact in a mesh-like structure around the spindles after cold treatment (Fig. S5E).

Fig. 7A and E: Consider providing a quantified data.

First revision

Author response to reviewers' comments

We thank the reviewers for their valuable time and efforts. We believe that in addressing the criticism raised during the review process, we were able to substantially improve the quality of our work.

Please find a detailed description of all changes in the accompanying point-by-point reply in blue. We have also submitted a manuscript version with changes highlighted in yellow.

Reviewer 1:

This study by Sajwan and colleagues investigates the non-transcriptional roles of Nub/Pdm1 in mitotic spindle assembly and dynamics. The topic is both timely and challenging, as it requires disentangling potential indirect transcriptional effects from direct, transcription-independent functions.

The manuscript presents a large body of data and employs several innovative tools. However, some of the findings are correlative and do not directly support the central claims of the paper. These issues could be addressed either by additional experiments or by significantly tempering the conclusions, or by limiting the number of experiments shown. I think the authors should present fewer things but present them better with the right controls. The multiplication of experiments in different systems (particularly in cultured cells) where the effect on transcription cannot be excluded does not necessarily help the message they want to get across.

Despite these concerns, I believe that the work has potential and could be suitable for publication in Journal of Cell Science after major revision and rewriting.

Major points:

* The use of deGrad-GFP or RNAi in embryos relies on maternal drivers. The authors argue that transcripts are only translated after egg activation, but this is an oversimplification. Disrupting mitosis during embryogenesis does not preclude prior effects during oogenesis, especially on the production of mitotic components by a known transcription factor. Therefore, many of the experimental strategies used may not be fully sufficient to draw conclusions about direct mitotic roles. The authors should adopt a more cautious interpretation to avoid misleading statements. In my view, the most convincing strategy is the antibody injection during syncytial divisions, which allows a rapid and stage-specific depletion. These experiments support the hypothesis of a direct role for Nub in spindle assembly and are a strength of the study.

We thank the reviewer for the comments. We agree, that in general, maternal driver-mediated depletion strategies may not fully exclude effects during oogenesis. This was also one reason for using several different approaches. We agree that the antibody injection is the strongest evidence for that the phenotypes are due to non-transcriptional function(s), and we have moved these results to Fig 5 where they contribute better to our conclusion. In respect to Nub protein, we have strong indications that it is not expressed in the oocyte. Immunostaining using two different antibodies show that Nub protein is undetectable in oocytes, and that it only appears after egg activation (Fig. S2C). Similarly, Nub-PD-GFP knock-in is not detectable during oogenesis, which render it unlikely that deGrad-GFP (or RNAi) degradation would influence mitotic components during oogenesis.

* Figure 1 and line 140: How can the mitotic index be decreased while metaphase/mitosis is delayed? These findings is contradictory and should be clarified.

We agree with the reviewer that this was not very clear. We believe that the discrepancy in our finding arises from the fact that we calculated the mitotic index based on PH3 staining, which labels late G2 and early mitosis but is lost during anaphase B and telophase. In S2 cells the predominant phenotype upon loss of Nub is multinucleation, indicating cytokinesis failure. It is therefore possible that these cells were missed in our quantification and consequently under-represented. For these reasons, we have decided to remove this panel.

* The efficiency of RNAi knockdown is not demonstrated. No Western blots are shown to validate depletion.

o Line 224: The absence of staining does not necessarily indicate the absence of protein. Have the authors performed Western blots to support this?

We acknowledge the lack of Western blots to demonstrate the RNAi efficiency. This has technical reasons. It has been a continuous frustration in my lab for more than two decades that it is so difficult to run Western blots for Nub-PD. It is not a problem of the antibodies, and we have used three types of antibodies: rabbit polyclonal poly-epitope antibodies (made against a large portion of Nub-PD protein), rabbit polyclonal peptide antibodies, as well as mouse monoclonal antibodies against Nub, and all give good immunostaining in tissues. Even if epitopes can be hidden by posttranslational modification(s) or present only in the native protein, we do not think this is the major cause of the inability to detect Nub-PD on Western blots from tissues/embryos. In fact, we detect Nub-PD on Westerns from in vitro translation experiments, from bacterially expressed protein, and by over-expression of Myc-tagged Nub-PD in cultured cells (Figure 1, below) supporting that the problem to detect Nub-PD is not due to the antibodies. Instead, we find that Nub-PD shows very poor extractability/stability when extracted from tissues and do not give bands on Westerns although immunostaining is prominent in those tissues with the same antibodies. We provide one Western blot further below (Figure 5) with extracts from wt and *nub*¹, to confirm the loss of Nub-PD in *nub*¹ embryos. We have previously also published Western blots with Nub-PD and Nub-PB from dissected midguts, where Nub is strongly expressed in enterocytes (see Additional File 2 in Dantoft et al, 2013, PMID: 24010524), but the quantitative reproducibility between experiments was poor also in those experiments.

After receiving the request for Westerns by this reviewer, we decided to give it another try. We have used a number of different extraction protocols and from several tissues, and managed to

see the expected band on a few blots but again, not with good quantitative reproducibility. Our lab uses Western blots as a routine for many other proteins, so this is not due to methodological inability. Thus, with the poor extractability/stability of Nub-PD protein it is not scientifically sound to use Western blots to judge the efficiency of RNAi.

Finally, we are not claiming that the RNAi is complete and we show in fact that it is partial, using mRNA quantifications (Fig S1C). We suggest instead, that both *nub* RNAi and the *nub*¹ mutant act as hypomorphs respectively, and that combining RNAi in the *nub*¹ mutant background leads to stronger and earlier effects on mitosis (Fig S5E-I).

Figure 1. Western blot analysis of Nub-PD-Myc overexpressed in *Drosophila* S2 cells. Anti-Myc (left) and anti-Nub (right) antibodies detect a band at the expected molecular weight for Myc-tagged Nub-PD.

* Line 252: The claim of defective chromosome segregation should be supported by appropriate live imaging, which is not convincing at all (e.g., Figure 2D). This is strange the authors decided to show these panels as nice movies are provided.

We had included a video showing this. See Video S3 and further description below under "Line 314 and 548".

* Line 352: What is the evidence for spindle "degradation"? Is this term appropriate?

We agree with the reviewer and have revised to "spindle MT de-polymerization".

* Line 308: Why would spindle dynamics be abnormal in controls?

Spindle dynamics are theoretically expected to be normal in wild-type controls. However, syncytial divisions are extremely rapid and highly sensitive to environmental perturbations. During live imaging in halocarbon oil, suboptimal gas exchange can lead to transient hypoxia, which is known to slow microtubule turnover and alter spindle behavior. In addition, repeated laser scanning can induce phototoxic effects and microtubule damage even in the absence of visible bleaching. These stresses are inherent to long-term high-resolution imaging and may occasionally result in subtle spindle abnormalities in control embryos. However, such events were rare in our work (<6% of analyzed embryos) and did not influence the overall conclusions.

* Line 314 and 548: Centrosome detachment from the spindle is a known defect, but is it the spindle that detaches from centrosomes or vice versa? Please clarify.

As we stated in the text and showed in Video S6, our live-imaging analyses indicate that spindle microtubules first detach from chromosomes, and not from centrosomes, and

subsequently attach to neighboring centrosomes, generating multipolar spindle arrays. These attachments become unstable and spindle bundles collapse and move inwards into the interior of the embryo, away from the cortex. However, centrosomes remain at the cortex in the same location. Although it is challenging to definitively determine whether the spindle releases from the centrosome or vice versa, we do not favor a centrosome-initiated detachment. We have revised the text to clarify this better.

* As mentioned above, conclusions from the deGrad-GFP phenotype are only valid if Nub-sfGFP is not degraded during oogenesis (It should control transcription of essential genes). A Western blot of Nub- sfGFP-D with and without the degradation driver during oogenesis would help determine whether the observed effects are truly embryo-specific. This control would also support better the antibody injection experiments and the "transcription independent effect" on mitosis.

The suggested experiment with a Western blot of Nub-sfGFP-PD with and without the degradation driver during oogenesis, would not be relevant as the nos-deGrad-FP and hb-deGrad-FP mRNAs are not translated in the oocyte, but upon egg activation/ovulation, and were previously shown to become active in the early embryo (Gaskill, 2021 (PMID 33720012) and Vazquez-Pianzola, 2022(PMID 35723263), see for example Figure 2 in Vazques-Pianzola. Thus, there is no target degradation in the oocyte.

Our results, as well as those from others, indicate that *nub-RD* mRNA is maternally transcribed (Figure 2, below) and mRNA is provided to the oocyte, and the mRNA is translated to Nub-PD protein during egg activation and oviposition (see figure S2C-E) and thereby, NubPD protein expression is only initiated in the syncytial embryo and then continues throughout embryogenesis (Figure 3, below). We have not been able to detect either Nub-PD or sfGFP-Nub-PD during oogenesis in germline stem cells (GSCs) or follicle stem cells (FSCs) by immunostaining, and we argue therefore that Nub protein is not translated/expressed in oocytes. We have observed some transient staining of a few centripetal follicle cells at late stage 10 (Figure 3 below), confirming that the antibody is working for staining of oogenesis, but there would not be any transfer of protein from these follicle cells to the oocyte. This result is supported by recent *Drosophila* ovary proteomics datasets, in which Nubbin protein was not detected.

Together, this supports that the observed effects are truly embryo-specific and are due to the loss of Nub- PD protein in the syncytial embryo.

Figure 2. Image from publicly available *Drosophila* scRNA-seq database (Fly Cell Atlas) shows that nubbin transcripts are detected in ovarian cells.

https://scope.aertslab.org/#/FlyCellAtlas/FlyCellAtlas%2Fee0d3b0a_20200415000Slaidina_et_al_2020_Larvae_Ovary.Harmony.loom/gene

Figure 3. sfGFP-Nub-PD expression during oogenesis and early embryogenesis. sfGFP-Nub-PD fluorescence is absent from the oocyte and germline cells during oogenesis, indicating that Nub protein is not maternally deposited. A transient signal is observed only in a few centripetal follicle cells at late stage 10 of oogenesis (arrow), in addition to some non-specific fluorescence in the vitelline membrane, present also in w1118 embryos. Robust nuclear sfGFP-Nub-PD protein expression becomes detectable during early embryogenesis and then continues in the expected pattern throughout embryogenesis (right part of the panel).

* Is the Nub-GFP-PD construct functional? Does it rescue the *nub1* phenotype? It remains unclear whether the *nub1* phenotype is due to loss of PB or PD isoforms. This distinction should be clarified, particularly if the authors aim to assign a specific role to Nub-PD in the MT regrowth assay.

It is not possible to introduce sfGFP-NubPD in *nub1* mutant background as a "rescue", as these are genetic changes in the same locus, and the sfGFP-NubPD is a knock-in in the *nub* locus, expressing sfGFP-NubPD from the promoter and enhancers deleted in *nub1*. (see also description of the *nub1* mutant below).

We conducted additional assays to directly test the functionality of sfGFP-NubPD, and we show that homozygous sfGFP-NubPD or heterozygous sfGFP-NubPD/ *nub1* animals have the same "relative nuclear to cytoplasmic ratio" in embryo nuclear cycles 11-12. In addition, the "total life span" of sfGFP-NubPD and sfGFP-NubPB is at wild type levels. In contrast, *nub1* shows strong relative N/C phenotypes and shorter life span (see figures below), indicating that sfGFP-NubPD confers full Nub-PD functions both in early embryos and during later life cycle stages. In revising the text, we explicitly clarify this and discuss the contribution of this specific isoform to the observed phenotype. The new data are shown in Fig. S5A-B in the manuscript and as Figure 4 below.

Figure 4. A. Relative nuclear to cytoplasmic ratio in embryo nuclear cycles 11-12 in corresponding genotypes. **B.** Life span analysis of sfGFP-NubPD flies in comparison to *nub1*, sfGFP-NubPB and w1118.

Response on the *nub1* phenotype: It is well known that the *nub1* mutant carries a transposon 412 insertion in the promoter region of *nub-RD* (PMID: 7768195) and a small deletion of a wing enhancer between *nub* and *pdm2* (PMID: 35853455). We have recently confirmed this by DNA

sequencing and by complementation analyses using null alleles for Nub-PD and Nub-PB (non-sense point mutations and deletions) in combination with *nub¹* (data not included in this manuscript). There are no indications that *nub*-RB expression is affected by the *nub¹* mutation. It is not impossible, however, that loss of Nub-PD could have a secondary effect on Nub-PB expression, but as shown in Figure S2B, only *nub*RD mRNA is expressed in early embryos (0-2 hr) and *nub*-RB is not expressed until late embryogenesis, just before hatching (by immunostaining using Nub-PB-specific antibody and by sfGFP-NubPB imaging, data not shown here). Consistently, we detected sfGFP-NubPD in early embryos but no sfGFP-NubPB signal, see Figure 5C-D in the manuscript.

Also see the Western blot below (Figure 5), where Nub-PD is present in wild type (*w¹¹¹⁸*) embryos and lost in *nub¹* embryos, and there is a very weak band for Nub-PB in this collection of embryos of all stages, while both isoforms are clearly present in S2 cells and wing discs. Overall, we have demonstrated that Nub-PD is present in syncytial embryos, but not Nub-PB, and plays a mitotic-specific role.

Figure 5. Detection of Nub-PB and Nub-PD isoforms in different *Drosophila* samples by Western blot. Lysates from *nub¹* embryos, wing discs, S2 cells, and *w¹¹¹⁸* embryos were probed with anti-Nub. β-Actin was used as a loading control.

* Figure 8, line 522: It is well known that altering spindle morphology or MT density (e.g., via nocodazole or colchicine) can affect the localization of MT-associated proteins. For instance, *Incenp* RNAi disrupts the spindle, so reduced Nub localization is expected. These measured effects may be indirect and do not demonstrate direct regulation of Nub by CPC. The authors could assess the Nub:MT signal ratio to strengthen this argument, but given the weak Nub spindle association and poor tubulin staining in some images, this might be technically challenging. I suggest removing the speculative recruitment model in Figure 8E and significantly softening the conclusions drawn from these experiments (see also discussion, lines 568-572).

We agree with the reviewer regarding the CPC complex. Although we also found conditions such as *AurA* or *Cdk1* knockdown that disrupt mitosis but leave Nub signal largely intact, CPC depletion broadly disrupts spindle architecture, and the resulting severe loss of Nub localization is therefore difficult to interpret as a direct, CPC-specific effect. We agree that this phenotype may be due to a general failure of mitotic structure rather than by a specific CPC-dependent regulatory mechanism. We now state more clearly that the reduced Nub signal after *Incenp* or other CPC component depletion likely reflects indirect effects of severe spindle disruption and compromised mitosis, rather than direct CPC-dependent recruitment.

In contrast, we need to point out that in several other RNAi conditions (like *Klp61F*, *Klp3A*, *Crb*, *Niki*, *Ned2*), spindle formation and mitotic progression remained normal, while Nub loss was specifically observed at metaphase and not at prophase. In these RNAi cases the reduction of Nub-PD levels was only partial (Table S1), and did not seem to fall below a threshold sufficient to disrupt mitotic progression. This is consistent with our observations in *nub1/+* heterozygous embryos (Fig. S5A), where Nub-PD signal is reduced but mitosis proceeds normal. Similarly, we observed that embryos from heterozygous mothers carrying the *Klp61F* or *Klp10A* mutation, where nuclear divisions are not detectably impaired, showed reduced Nub only in *Klp61F/+* but

not in Klp10A/+ spindles. This indicates that Nub association with spindle microtubules is highly dosage-sensitive and depends on “specific” regulatory cues, rather than being a passive consequence of general spindle perturbation. Together, these observations suggest that spindle localization of Nub is regulated during metaphase and is not merely an indirect consequence of a general mitotic or spindle perturbation. We have incorporated these clarifications into the Results section. The CPC-specific interpretation has been removed from the schematic, while the indicative effects of the other mitotic components shown in the graph. Figure 8E has been revised accordingly.

Finally, we agree that the tubulin staining is challenging. We used a more robust internal normalization strategy and quantified Nub-PD intensity within the spindle region relative to adjacent non-spindle regions in the same nuclei/embryo. This approach allows us to compare the relative enrichment of Nub on spindles across conditions in a more reliable manner (Table S1).

* There is confusion regarding Nub localization “around the spindle.” Given that the spindle is compartmentalized by membranes (as shown in works by Maiato, Baum, and others), the current terminology is misleading. In embryos, Nub1 appears around chromosomes, whereas in S2 cells, it co-localizes with the spindle.

We agree and we have revised the text to state that Nub localizes within the spindle region enclosed by these membranes or within the “spindle envelope” as defined by the Schweizer et al. 2015 (JCB). In S2 cells, Nub co-localizes more directly with spindle microtubules.

In syncytial embryos the common cytoplasm might induce condensates or puncta (dots/mesh) of Nub rather than uniformly coating structures, due to local phase separation and cytoplasmic flows. We have included some discussion on the possibility that Nub-PD functions by acting within condensates on microtubules.

In S2 culture cells spindles are compartmentalized within an individual cell with a well-defined cortex, making localization more continuous and symmetric, so Nub appears as a spindle “coat” rather than discrete puncta. S2 cells are also dividing more slowly, may allow Nub to establish a more stable equilibrium with spindle MTs and MAPs.

* The interpretation of the MT regrowth assay is unclear and should be more discussed.

We used the cold-induced MT regrowth assay to examine how embryos and cultured cells rebuild a functional bipolar spindle after acute microtubule loss, and to determine the contribution of Nub-PD and POU2F1/Oct1 to this process. We added a more distinct interpretation in the discussion on this.

* All figures and movies must include time stamps and scale bars. This is a minimal requirement for publication in a cell biology journal, yet many figures lack this.

We thank the reviewer for this notification. We have revised the figures.

* Fixation may mask epitopes. Could the authors provide live imaging movies of sfGFP-Nub isoforms with RFP- or mCherry-tubulin?

Unfortunately, we cannot detect efficient signal of sfGFP-NubPD by live imaging in the early syncytial embryos. Therefore, we focused our analysis on fixation conditions at these earlier stages.

We also tested multiple fixation conditions, including as low as 0.5% PFA, and Nub localization remained unchanged in the spindles. If chemical fixation was compromising Nub signal, we would expect a loss of the strong chromosomal Nub signal in prophase, which we do not observe (Fig. 6 and S6). Therefore, we consider the fixation conditions appropriate and not responsible for the observed Nub localization phenotypes.

* S2 cells (Figure S1 C-H): S2 cells express both isoforms, but RNAi only achieves ~50% knockdown of the 2 (B and D) RNAs. No Western blots are provided, which raises concerns about the reliability of the reported experiments. Additionally, I am surprised that the authors use midbody staining in cells at various stages of division (from telophase to abscission) to assess protein

depletion. This approach is unconventional and not standard practice in the field.

Please see our discussion above on the technical problems with using Western blots to measure Nub-PD RNAi, and our data that shows that Nub-PD but not Nub-PB is expressed in early embryos. We used the midbody staining in S2 cells to show that Nub-PD RNAi, but not Nub-PB RNAi, affected antibody staining during mitosis and this was our first indication that Nub-PD is responsible for functions during mitosis (since the antibody recognizes both isoforms). But we agree with the reviewer that these data can be confusing in the present context where we also have much more convincing results, and we have removed those panels.

* Figure S1 I-L: The authors suggest Nub function is conserved in human cells, but the phenotypes are poorly characterized. Multipolar spindles may result from centriole amplification or fragmentation, whereas multinucleation (as indicated in the graphs) implies cytokinesis failure. No images of interphase polyploid cells are shown to sustain that hypothesis.

We agree with the reviewer. In general, centriole amplification or fragmentation can lead to multipolar spindle formation, whereas multinucleation may arise from cytokinesis defects in human cells. However, our study did not assess these mechanisms directly, and therefore we cannot conclude which process underlies the observed phenotypes. In *Drosophila* embryos, our analysis indicates that centrosome number is not altered and cytokinesis does not occur during early syncytial divisions, making spindle collapse the most possible cause of nuclear mitotic defects. In contrast, both scenarios remain possible in human cells, and further experiments will be required to distinguish between centrosome-related defects and cytokinesis failure in this context. To avoid overinterpretation, we have brought this point into the discussion.

* The MT regrowth assay suggests that chromosome-driven MT nucleation (regulated by Ran/CPC) is affected by Nub. As Nub contains an NLS, it may be relevant to discuss possible connections with Ran- dependent pathways. Interestingly, centrosome-based MT nucleation increases in Nub mutants during anaphase, while central spindle MTs decrease. Can the authors explain this phenotype?

We thank the reviewer for this comment. Our data from *Drosophila* embryos indicate that Nub-PD is not strictly required for centrosome-based MT polymerization, as regrowth from centrosomes is still observed and chromosome-proximal MTs can reform in many *nub¹* embryos with same fluorescence intensities to controls (Fig. 5, below). Instead, the predominant defect lies in the failure to convert regrowing MTs into a coherent bipolar spindle with a robust interpolar midzone. We therefore propose that Nub-PD acts as a regulator of spindle reassembly and stability, promoting the bundling, organization and stabilization of regrowing MTs. In addition to this organizational role, Nub-PD may also contribute directly or indirectly to efficient reactivation and/or amplification of MT polymerization after acute depolymerization, as suggested by the complete lack of MT regrowth in a subset of *nub¹* embryos. We do not know the mechanism. A link to Ran/CPC-dependent chromatin-based nucleation is possible and intriguing but remains speculative at present. We have mentioned this in the result section now.

In *nub¹* mutant embryos, we indeed observed a dominant centrosome-based MT enrichment during recovery. We do not think that centrosome-based MT nucleation increases *per se*. Live imaging analysis shows that both centrosomal and chromatin-directed MT polymerization are initiated normally (Figure 6, below), but MTs extending from the centrosomes toward the chromosomes fail to be maintained in the central part of the spindle and instead persist as elongated astral-like arrays clustered around the centrosomes (Figure 7D, manuscript). This is accompanied by a reduction of central spindle MTs in pre- anaphase B. These phenotypes are consistent again with the spindle organization defects observed in *nub¹* mutants under normal conditions, supporting our conclusion that Nub is crucial for maintaining bipolar spindle stability.

Figure 6. Images showing wild type embryos and *nub*¹ embryos expressing EB1-GFP after cold treatment.

The chromatin-directed MT polymerization is indicated by arrows.

* Figures 6 and 7 appear somewhat redundant. Is the function of human Oct1 different from that of Nub? The phenotypes differ. This should be addressed.

We agree that the phenotypes are different. This difference could reflect system-specific biology or perhaps arise from the broader transcriptional roles of OCT1 in human cells, which may generate additional phenotypes not present in *Drosophila* embryos. OCT-1 localizes to the midbody in human cells, suggesting a potential cytokinetic function that is absent in syncytial *Drosophila* embryos, where cytokinesis does not occur. We have added this argument into the text.

* Figure 5C-D: It is unclear whether these panels represent different treatments. Please clarify in the legend.

These panels do not represent treatments. They show the expression of sfGFP-NubPD and sfGFP- NubPB in two NCs of the early embryo development. We have revised the labeling.

* Figure 3: Pole-to-pole distance decreases in Nub1 mutants at metaphase in cycle 9 but remains unchanged during pre-anaphase B in cycle 10. Is the phenotype cycle-specific?

We apologize for the unclarity. In NC9, we measured spindle length at the initiation of anaphase B (Fig. 4G), which correlates with the differences shown in the graphs of NC10 and NC11 (Figure 4I-J). We have clarified this for NC9 measurements in the new version and in figure legends.

* Figure 2D: On what basis do the authors claim these are anaphase figures in Nub1 mutants? From the others?

Please see below.

* Figure 2E-G: The measurement of cell cycle phases is surprising given the poor image quality. I would therefore suggest to show better images with a higher magnification. NEBD cannot be assessed without nuclear envelope markers. Alternatively, nuclear entry of cytoplasmic proteins in the nuclear space could be used. Without this, the measurements are unreliable. I strongly recommend showing only data supported by the H2B nuclear marker (if that is what was used). Otherwise, these data should be removed. At least the authors could explain how the data were obtained in the material and Method section.

Data on nuclear division phases are supported by the H2B nuclear marker. We removed the panel 2G which reported NEBD measurements, as this assessment had been conducted without the use of a specific nuclear envelope marker and could not be interpreted with sufficient confidence. We provide a new panel in Figure 2, showing the chromosome segregation defects in *nub*¹ and MTD>nub RNAi among mitotic phases (Fig. 2D-I, and see a copy below as Figure 7). In addition, we have added a new panel in Figure 3A (see copy below as Figure 8) to clarify how S-phase and M-phase were defined in our analysis. Details about the measurement of cell cycle phases are included in the Material and Methods section.

Figure 7 (D-H) Representative confocal images of *Drosophila* pre-blastoderm embryos at metaphase (D), anaphase (F), and telophase (H) stained with DAPI to visualize chromosomes. Wild-type embryos show properly aligned metaphase chromosomes (yellow arrowheads), normal segregation during anaphase (purple arrowheads), and evenly sized daughter nuclei in telophase (green arrowheads). In contrast, *nub*¹ mutant and *MTD>nub* RNAi embryos display chromosome misalignment, lagging chromosomes, and unequal chromosome distribution, indicating disrupted mitotic progression. (Fig 2 D-I in manuscript)

Figure 8 (A) Reference panel showing DAPI-stained nuclei from *Drosophila* syncytial embryos, used to define S-phase (interphase to late prophase) and M-phase (metaphase to late telophase) during nuclear cycles (NC10-13). (Figure 3A in manuscript)

* Figure 1I: What exactly is plotted? How are values normalized?

Each point represents average values of pole-to-pole distance over time in two nuclear cycles. In the figure legends, we write the number of embryos analyzed in each time point.

* No Western blots are shown to validate the new anti-Pdm2 antibodies.

We appreciate this comment. We now include a Western blot from wild-type embryos showing the two expected Pdm2 isoforms, confirming that the antibody recognizes endogenous Pdm2 protein. In early embryos, we further demonstrate specificity by comparing staining after GFP RNAi versus pdm2 RNAi. Upon pdm2 depletion, the Pdm2 signal is completely abolished from its characteristic chromosomal localization, whereas GFP RNAi has no effect. Together, these results validate the specificity and reliability of the new anti-Pdm2 antibodies.

Figure 9. Left panel: WB showing the Pdm2-PA and Pdm2-PB isoforms of wild type embryos. Right panel: staining of Pdm2 antibody in GFP RNAi and pdm2 RNAi knockdown embryos using the maternal MTD-GAL4 driver. (Figure S6D-E in manuscript)

Unsupported claims:

* Line 523: "This spindle enrichment is microtubule-dependent." No data are shown to support this.

We refer to the cold-induced depolymerization experiment, where Nubbin is no longer detectable on depolymerized spindles, supporting its association with intact microtubules (Figure 6F). Upon recovery, both microtubules and Nubbin spindle localization recover. This reversible loss specifically upon microtubule depolymerization demonstrates that Nubbin enrichment at the spindle requires intact spindle microtubules. We have revised to "This enrichment requires intact microtubules" in the text to tone down our statement.

* Line 532: "...and thereby for its mitotic functions." This is not demonstrated.

It is an assumption. Nubbin localizes properly in spindles and we assume that is functional as we did not observe any mitotic defects. We change the text as: and likely for its mitotic functions as we did not observe mitotic defects.

* Line 543: "Thus, Nub-PD and POU2F1/Oct1 play dual roles..." This is speculative for the human ortholog. As the authors themselves state in line 609-610, "Whether this mitotic role of POU2F1/Oct1 is mechanistically distinct from its transcriptional activity remains to be determined."

The phenotypes strongly support a role of both Nub-PD and POU2F1/Oct1 in mitosis. This also with agreement with earlier work. We propose that POU transcription factors may have dual functions: in interphase, they can act as general transcriptional regulators controlling programs such as cell fate specification, stress responses and immunity, whereas during mitosis they can function as regulators of spindle organization and timely mitotic progression. From our work in *Drosophila* syncytial embryos, where zygotic transcription is absent, we obtain direct evidence for a non-transcriptional role of Nub-PD in mitosis. In human cells, however, the situation is less clear, and POU2F1/Oct1 could contribute to mitotic fidelity through non-transcriptional

mechanisms, through transcriptional programs established already at the interphase, or through a combination of both. Our work cannot answer whether this mitotic role is transcriptional or non-transcriptional or both.

* Line 547: "Our data show that both centrosome numbers and morphology remain unaffected." Yet, centrosome numbers are not quantified. Multipolarity could result from cytokinesis defects or centriole amplification. Please amend this sentence or provide data.

We agree with reviewer. We provide the quantifications in the new version. Our quantification indicates that centrosome numbers are not detectably altered in *nub¹ Drosophila* embryos, suggesting that the observed multipolar spindles are unlikely to arise from centrosome amplification. This further supports our conclusions that Nub functions at the level of the spindle rather than the centrosomes, consistent with its absence of centrosomal localization.

* Line 558: "...supporting proper spindle architecture and dynamics during metaphase and anaphase." Why not earlier mitotic stages?

We specifically refer to metaphase and anaphase because these are the stages where we clearly see defects in our data. Live imaging and immunostainings show abnormal spindle morphology and dynamics, as well as chromosome segregation errors, predominantly from metaphase onwards. In contrast, we do not detect obvious defects at earlier stages: a) PH3 staining does not reveal abnormal chromosome condensation, and 2) we do not observe S-phase delays or impaired mitotic entry in our time lapse. To avoid overextending our conclusions beyond what is supported by the experiments, we therefore restrict our statement/conclusion to metaphase and anaphase.

* Line 600: "...and the formation of multinucleated cells." Please provide images or FACS analysis to support this, or remove the claim.

We thank the reviewer for this comment. We now support this statement with additional data. We performed staining of Lamin-B in S2 cells showing the presence of multinucleated cells in Nub-PD knockdown (new panel in Figure S1 and Figure 9 below). These data confirms that loss of Nub-PD can result in defective nuclear division and multinucleation.

Figure 10. Lamin-B staining in S2 cells.

* Line 604: "...promoting microtubule regrowth in human cells." This is not evident from Figure 7E, which is also not quantified.

We provide now the quantitation. We have repeated the cold-induced microtubule depolymerization assay in HeLa cells with increased sample size and quantified: (i) spindle microtubule coherency during recovery, and (ii) the percentage of fully reformed spindles over time in control siRNA versus Oct1 siRNA-treated cells. Both analyses show a significant delay and reduction in spindle regrowth upon Oct1 depletion, suggesting that Oct1 is required for efficient and timely microtubule recovery after depolymerization. We have incorporated these data into the

revised Figure 7, added details in M&M section and updated the text accordingly.

The data does not reflect a role on mitotic progression (as mentioned in the title) but on spindle assembly and chromosome segregation.

We propose a new title: **A Non-Transcriptional Mitotic Function of POU/Oct Factors Ensures Spindle Stability and Chromosome Segregation.**

Reviewer 2: This study revealed that *Drosophila* Nub1/Pdm1 and its human homolog POU2F1/OCT1 play a role in ensuring accurate mitosis, which is distinct from their well-known function in transcription.

Nub1 has two isoforms, Nub1-PB and Nub1-PD in *Drosophila*. siRNA-mediated knockdown of Nub1-PD caused deformation of the mitotic spindle in *Drosophila* S2 cells (Fig. 1A-C). Similarly, siRNA against OCT1 (siOCT1) induced mitotic defects in human HeLa cells (Fig. 1D-E). In *Drosophila* embryos, only Nub1-PD is expressed, and zygotic transcription is not required for the syncytial division cycles. Under these conditions, nub1 mutants exhibited mitotic defects, and the embryo cortex was not fully covered by nuclei (NUF phenotype; Fig. 2). As observed in S2 cells, nub1 mutant embryos showed disorganized mitotic spindles, prolonged mitosis, and shortened spindle length (Fig. 3). The NUF phenotype and spindle defects were rescued by introducing an additional copy of the nub1 gene (Fig. 4). Protein knockdown of GFP-fused Nub1-PD using deGradFP caused even more severe mitotic defects in embryos. Nub1-PD localized around chromosomes and spindle bundles during metaphase in a tubulin-dependent manner (Fig. 6). The spindle localization of Nub1-PD depends on Nek2, Niki, the CPC complex, Klp61F, Klp3A, and Crb (Fig. 7).

Overall, the study provides convincing evidence that Nub1-PD plays a role in proper mitotic spindle formation, which is distinct from its function in transcription. I did not find a major problem for publication. However, the authors should consider the following minor points in a revised version.

Line 156: Fig. SG-H should be corrected to Figure S1G-H.

We thank the reviewer for noticing this. It has now been corrected to Figure S1G-H.

Fig. 5E-G: the presented images show clear defect in mitosis at NC9. However, it is more convincing to include a quantified data as well. Additionally, is it possible to observe the defect at earlier NC stages?

We thank the reviewer for this suggestion. We have repeated the experiment using *hb>deGradFP* and collected additional data for quantification, which are now included in the revised Figure 5 (panel 5E-F). These results further support the efficiency of the deGradFP system and strengthen the conclusion that Nub is required for proper mitosis in early nuclear divisions. The mitotic defects are already detectable at earlier nuclear cycles, which confirms that the phenotype is not restricted to NC9.

We observe defects in earlier stages. However, prior to NC9, the nuclei are located deeper within the embryo, and the high yolk content causes strong light scattering. Even with confocal Airyscan acquisition, these optical limitations prevent reliable visualization of spindle structures or chromosome arrangement and do not allow accurate quantitative analysis with sufficient resolution or signal-to-noise. For this reason, we based our quantification on NC9 and onward, where the nuclei are positioned at the cortex and can be imaged robustly.

Lines 424-427: Megator showed a similar localization pattern as Nub-PD in wild type embryos, and this did not change in nub1 mutant embryos. In contrast to Nub-PD, Megator remained intact in a mesh-like structure around the spindles after cold treatment (Fig. S5D). These sentences should be as follows.

Megator showed a similar localization pattern as Nub-PD in wild type embryos, and this did not change in nub1 mutant embryos (Fig. S6F). In contrast to Nub-PD, Megator remained intact in a mesh-like structure around the spindles after cold treatment (Fig. S6G).

We have corrected this. Thank you.

Fig. 7A and E: Consider providing a quantified data.

We now provide new quantified data showing centrosomal and spindle tubulin intensity in cold-treated embryos, in embryos allowed to recover, and in untreated controls (Fig. 7B and Fig. 7C). In addition, we quantified spindle microtubule coherency over the recovery period, as well as the percentage of reformed spindles in control siRNA and Oct1 siRNA-treated human cells (Fig. 7G and Fig. 7H). Both quantifications show that Nubbin and Oct1 are required for proper and timely microtubule regrowth.

Second decision letter

MS ID#: jcs.264165R1

MS TITLE: A Non-Transcriptional Mitotic Function of POU/Oct Factors Ensures Spindle Stability and Chromosome Segregation

AUTHORS: Priya Gohel; Vasilios Tsarouhas; Laveena Kansara; Suresh Sajwan; Ylva Engström

ARTICLE TYPE: Research Article

Dear Dr Engström,

We have now reached a decision on the above manuscript.

As you will see, the reviewers gave favourable reports. One of the reviewers suggests a few modifications that will require amendments to your manuscript. I hope that you will be able to carry these out because I would like to be able to accept your paper very soon.

Reviewer 1

Advance summary and potential significance to field

The authors have addressed most of my concerns. Given the work produced in this revised study and the efforts made in writing and presentation, I feel that the paper is suitable for JCS.

I would suggest to replace "stability" by "assembly" in the title.

However, I have noted some minor points that nevertheless need to be amended, as well as a few suggestions. I don't think I need to review this manuscript again.

L233: "suggesting that Nub-PD has a function in the very first and all subsequent nuclear divisions of the syncytial embryo." This statement is somewhat of a shortcut; please amend.

As far as I understand, S-phase duration is measured by DAPI staining. I am wondering whether this method is suitable for an accurate measurement of S phase. I would suggest including relevant references to convince readers regarding the methodology. Alternatively, the data could be removed if the authors do not wish to distract readers from the main message.

Figure 7H: I am not sure about the "anaphase stage" of the presented cell after only a 5-minute regrowth, especially since no DNA staining or Cyclin B staining is shown. Please amend the scheme by showing only MTs and avoid referring to "anaphase" in the text.

I am still hesitant regarding the interpretation and writing of the last Results section (Figure 8). CDK1 is required for mitotic entry, and mitotic cells are examined here, suggesting that depletion is partial and not sufficient to prevent mitosis. Klp61F depletion is also known to produce monopolar

spindles in many systems, whereas bipolar spindles are clearly observed here. Altogether, I feel that the authors are working under moderate depletion conditions, and this should be mentioned.

L563: "Interestingly, we found that sequence-specific DNA binding and transcriptional regulation are not important for Nub-PD localization to the spindle, and likely not for its mitotic functions, as we did not observe any mitotic defects." This sentence seems incomplete.

L593: "that nub1 embryos can initiate MT regrowth around chromosomes." This regrowth appears to be strongly diminished in the embryos shown in Figure S6H. Please amend as this experiment suggest the chromatine nucleation pathways regulated by RAN and the CPC is affected.

L606: "nucleation, although this remains speculative at present. Nub-PD may be anchored in the spindle apparatus via other MT-associated proteins (MAPs)." Could this involve Klp61F ?

Reviewer 2

Advance summary and potential significance to field

The authors have revised the manuscript to a level sufficient for publication. I have no further comments, except to congratulate them on their work.

Second revision

Author response to reviewers' comments

Comments from the Reviewers, and responses in blue:

Reviewer 1: The authors have addressed most of my concerns. Given the work produced in this revised study and the efforts made in writing and presentation, I feel that the paper is suitable for JCS.

I would suggest to replace "stability" by "assembly" in the title.

We thank the reviewer for this suggestion, we have made this change.

However, I have noted some minor points that nevertheless need to be amended, as well as a few suggestions. I don't think I need to review this manuscript again.

L233: "suggesting that Nub-PD has a function in the very first and all subsequent nuclear divisions of the syncytial embryo." This statement is somewhat of a shortcut; please amend.

As far as I understand, S-phase duration is measured by DAPI staining. I am wondering whether this method is suitable for an accurate measurement of S phase. I would suggest including relevant references to convince readers regarding the methodology. Alternatively, the data could be removed if the authors do not wish to distract readers from the main message.

We agree with the reviewer that we cannot conclude functional requirement at this stage. We therefore have removed our statement and amended the text to indicate that Nub-PD is already translated and recruited to spindle microtubules from the first nuclear division (line 231-234).

Regarding S-phase timing, we did not quantify S phase from fixed DAPI measurements. Instead, S-phase/interphase duration was determined in live embryos using the His2A-RFP chromatin reporter and measured from post-mitotic chromatin de-condensation to the onset of visible chromosome condensation, following established methodology (Fasulo et al.2012 ; Yu et al., 2000, Nat Cell Biol). We have clarified this definition in the legends of Figure 3, and in the Methods and the Results sections to prevent any misunderstanding. We also included the corresponding references.

Figure 7H: I am not sure about the "anaphase stage" of the presented cell after only a 5-minute

regrowth, especially since no DNA staining or Cyclin B staining is shown. Please amend the scheme by showing only MTs and avoid referring to "anaphase" in the text.

We agree with the reviewer. We assume the reviewer is referring to Figure 7I and not to the figure 7H. We revised the schematic panel (7I) to display only microtubules (β -tubulin/MTs) in wt and *nub*¹ embryos or siRNA Oct1 human cells. We also removed the term "anaphase". Now the described graphical panel is focused on spindle reassembly/MT organization rather than mitotic staging.

I am still hesitant regarding the interpretation and writing of the last Results section (Figure 8). CDK1 is required for mitotic entry, and mitotic cells are examined here, suggesting that depletion is partial and not sufficient to prevent mitosis. Klp61F depletion is also known to produce monopolar spindles in many systems, whereas bipolar spindles are clearly observed here. Altogether, I feel that the authors are working under moderate depletion conditions, and this should be mentioned.

In the maternal knockdown screen (Fig. 8/Table S1), we aimed for moderate/partial depletion conditions so that embryos could continue through syncytial divisions and Nub-PD localization could be assessed in *bona fide* mitotic spindles. For some CPC components, depletion produced severe mitotic defects and severe Nub mis-localization across multiple mitotic phases, making it difficult to interpret Nub-specific effects; we therefore omitted these conditions from the proposed graphical panel in our previous resubmitted version.

As the reviewer pointed out, complete depletion of core mitotic regulators such as Cdk1 would be expected to block mitotic entry, and strong depletion of Klp61F typically yields severe spindle phenotypes (e.g., monopolar spindles). The presence of normal mitotic phases and largely bipolar spindles in our material indicates that the RNAi conditions were not fully penetrant, consistent with a "partial knockdown". Essentially, even under these moderate conditions, we consistently detected defects in Nub-PD spindle localization. Under stronger depletion conditions, the overall mitotic disruption was too severe to distinguish Nub-specific recruitment effects from general mitotic collapse. We acknowledge that this strategy may also reduce the sensitivity to detect additional localization phenotypes of Nub-PD.

We have now explicitly stated in the Results (lines 522-527) and in the Materials and Methods (lines 700-703) that these experiments were performed under moderate depletion conditions, selected to preserve overall mitotic progression and allow assessment of Nub-PD recruitment. We further clarify from our previous revision that for factors such as CPC components that disrupt mitotic progression, Nub mis-localization may reflect both direct effects and secondary consequences of impaired mitotic entry; therefore, these conditions are not shown in Fig. 8E. We instead present genes that specifically altered Nub-PD metaphase localization without broadly affecting mitotic progression, and interphase signals, consistent with a more specific role in spindle recruitment.

L563: "Interestingly, we found that sequence-specific DNA binding and transcriptional regulation are not important for Nub-PD localization to the spindle, and likely not for its mitotic functions, as we did not observe any mitotic defects." This sentence seems incomplete.

We have revised this sentence (lines: 559-562).

L593: "that *nub*1 embryos can initiate MT regrowth around chromosomes." This regrowth appears to be strongly diminished in the embryos shown in Figure S6H. Please amend as this experiment suggest the chromatin nucleation pathways regulated by RAN and the CPC is affected.

As requested by the reviewer, we have amended the text in our Discussion to reflect that the diminished/absent regrowth is possibly due to defects in chromatin-mediated microtubule nucleation and/or stabilization, potentially involving Ran- and/or CPC-dependent pathways (lines 599-603).

L606: "nucleation, although this remains speculative at present. Nub-PD may be anchored in the spindle apparatus via other MT-associated proteins (MAPs)." Could this involve Klp61F ?

Klp61F and Klp3A are indeed candidates to contribute to Nub-PD spindle association, since knockdown of both reduces Nub-PD enrichment specifically on metaphase spindles in our screen. However, we have not performed experiments that test this in a direct physical interaction or define the recruitment mechanism. To avoid over-interpretation (and because mechanistic dissection is beyond the scope of this manuscript), we have just added a cautious sentence in the Discussion noting that Nub-PD recruitment in the spindles could occur via Klp61F or Klp3A -or other MAPs, not identified in our study (lines 605-611).

Reviewer 2: The authors have revised the manuscript to a level sufficient for publication. I have no further comments, except to congratulate them on their work.

Thank you!

We additionally included quantifications of the mitotic phenotype upon depletion of sfGFP-Nub- PD using *nanos >deGradFP* flies (Fig. S5C''').

Third decision letter

MS ID#: jcs.264165R2

MS Title: A non-transcriptional mitotic function of POU/Oct factors ensures spindle assembly and chromosome segregation

Authors: Priya Gohel; Vasilios Tsarouhas; Laveena Kansara; Suresh Sajwan; Ylva Engström

Article Type: Research Article

Dear Dr Engström,

I am happy to tell you that your manuscript has been accepted for publication in Journal of Cell Science, pending standard publication integrity checks.